# Neural timescales reflect behavioral demands in freely moving rhesus macaques

Ana M. G. Manea [1,2] ✉, David J.-N. Maisson[1], Benjamin Voloh[1], Anna Zilverstand [3], Benjamin Hayden [4] & Jan Zimmermann [1,2]

Previous work demonstrated a highly reproducible cortical hierarchy of neural timescales at rest, with sensory areas displaying fast, and higher-order association areas displaying slower timescales. The question arises how such stable hierarchies give rise to adaptive behavior that requires flexible adjustment of temporal coding and integration demands. Potentially, this lack of variability in the hierarchical organization of neural timescales could reflect the structure of the laboratory contexts. We posit that unconstrained paradigms are ideal to test whether the dynamics of neural timescales reflect behavioral demands. Here we measured timescales of local field potential activity while male rhesus macaques foraged in an open space. We found a hierarchy of neural timescales that differs from previous work. Importantly, although the magnitude of neural timescales expanded with task engagement, the brain areas' relative position in the hierarchy was stable. Next, we demonstrated that the change in neural timescales is dynamic and contains functionally-relevant information, differentiating between similar events in terms of motor demands and associated reward. Finally, we demonstrated that brain areas are differentially affected by these behavioral demands. These results demonstrate that while the space of neural timescales is anatomically constrained, the observed hierarchical organization and magnitude is dependent on behavioral demands.

Behavioral coordination and adaptation across an ever-changing environment are a hallmark of cognition in biological systems. To function in our daily lives, we simultaneously consider auditory, visual, and sensory input while achieving motor coordination, each of which spans a continuum of spatial and temporal scales. Consider one of the most studied systems in neuroscience, the visual cortex. It is well known that neurons along the visual pathway have increasingly larger receptive fields[1]—higher-level visual areas respond to information from large portions of space by integrating input from neurons in the early visual cortex, which possess smaller receptive fields. Events not only unfold over multiple spatial but also over a multitude of temporal scales[2,3]. Indeed, this hierarchical increase in representational

complexity is closely followed by a hierarchy of longer temporal processing windows[4]. Similarly, neural processing in the prefrontal cortex (PFC) is organized across different scales of complexity, with progressively more abstract representations and higher-order control on posterior-anterior and ventral-dorsal axes[5–7], Moreover, electrophysiological and functional magnetic resonance imaging (fMRI) results in human and nonhuman primates at rest have demonstrated that the frontal lobe is organized along a hierarchical gradient of neural timescales that mirrors its functional architecture[8–11]. In particular, it is thought that areas that display relatively more information integration display slower neural timescales[12–15]. These parallel results suggest that neural timescales in the PFC might be functionally relevant,

[1]Department of Neuroscience, University of Minnesota, Minneapolis, MN, USA. [2]Center for Magnetic Resonance Research, University of Minnesota, Minneapolis, MN, USA. [3]Department of Psychiatry and Behavioral Sciences, University of Minnesota, Minneapolis, MN, USA. [4]Department of Neurosurgery, Baylor College of Medicine, Houston, TX, USA. ✉e-mail: manea006@umn.edu

nevertheless, direct evidence to support this conclusion is limited. From one perspective, it could be argued that neural timescales merely reflect a relatively static property inherited from their place in the anatomical hierarchy that allows neurons within that area to integrate over a stable temporal scale. In this case, although neural timescales would facilitate function, the timescales themselves would not change in a functional manner. Alternatively, it could be argued that anatomy instead might impose a range of timescales that bounds the dynamics over which brain areas can operate. For example, it is conceivable that there are stable hierarchies of timescales that can expand and contract depending on functional demands.

To answer this question, we need to investigate the dynamics of neural timescales in the context of behavior[8,16–18]. There is currently scarce evidence about the behavioral dependence of neural timescales, their hierarchical organization, and general function in the context of behavior, and especially how these relate to one another. Some preliminary evidence suggests that neural timescales expand with task engagement and attention, and hence are potentially functionally relevant[16,18]. In contrast, a plethora of previous findings have demonstrated remarkably reproducible and consistent neural timescales across cortical areas at rest (see Fig. 1A), with one study even concluding that the hierarchy of neural timescales appears invariant to task context and that neural timescales are not affected by behavioral demands[19]. In contrast, a large-scale dynamical model of the macaque neocortex exhibits not one, but multiple temporal hierarchies, as indicated by unique responses to visual and somatosensory stimulation[20]. The existence of multiple concurrent neural timescales gradients that are dynamically expressed[20,21], might accommodate adaptive and flexible behavioral changes at different temporal scales. Finally, although characteristic timescales have been assigned to brain areas as a whole, single neurons display heterogeneous neural timescales at rest[10,22–26]. The question arises whether this heterogeneity is purely anatomical or whether it is the result of both anatomy and contextual demands[27].

We propose that the failure to find multiple hierarchies of neural timescales in previous experiments, i.e., the areas' relative position, may be a by-product of the rigid structure of traditional experiments that enforce stationary temporal scales. Investigating neural timescales in a constrained lab environment (i.e., chaired electrophysiology) with trialized tasks, imposes a bounded temporal structure and limits the complexity of the input entering and the output leaving the brain. If these constraints were removed, neural and behavioral dynamics would become temporally unconstrained— except for the boundaries imposed by biophysics. Therefore, what we know about neural timescales might not be entirely intrinsic to the biological system, but rather a reflection of the conditions imposed by the structure of the experimental paradigm[28,29]. While this approach has brought invaluable contributions to our understanding of the brain and behavior, it is nonetheless limited when it comes to studying the functional relevance of neural timescales. To understand and establish a neurobehavioral timescale correspondence, it is thus imperative to approach the question from a more unconstrained perspective.

Here, we investigated neural timescales in an unconstrained experimental paradigm with relatively minimal temporal structure. We hypothesized that the observed hierarchy of neural timescales is dependent on the environment, as reflected by the temporal constraints, the complexity and nature of the input, and the required motor output. Moreover, we predicted behaviorally-dependent shifts in the magnitude of neural timescales. To test these hypotheses, we investigated how the brain handles multi-scale signals to drive purposeful behavior while rhesus macaques were free to forage in a large open field environment[30–32]. Our experimental paradigm imposed minimal temporal constraints and put emphasis on self-paced behavior rather than focusing on a particular cognitive or perceptual

process in isolation. We simultaneously recorded brain activity in eight brain areas: orbitofrontal cortex (OFC), ventrolateral prefrontal cortex (VLPFC), dorsolateral prefrontal cortex (DLPFC), frontal eye fields (FEF), anterior cingulate cortex (ACC), premotor (PM) cortex, supplementary motor area (SMA), and dorsal striatum. We investigated neural timescales from a population perspective as reflected in the local field potential (LFP) activity[16,33]. We found a hierarchy of neural timescales different from previous studies, potentially shaped by this foraging environment. Next, we demonstrated that neural timescales expand with task engagement, although the areas' relative position in the hierarchy remains the same across the recording session. Finally, we showed that the change in neural timescales is dynamic and reflects the abstract meaning of foraging events. Together, this demonstrates that while anatomy constrains the space of possible neural timescales, the observed hierarchical organization and magnitude of neural timescales is heavily dependent on behavioral demands.

## Results

Two macaques performed a foraging task in a large open space that allowed for unconstrained movement (Fig. 1B and Methods). The environment contained four reward stations positioned at fixed locations. The reward stations dispensed 1.5 mL of liquid reward for each of the first four lever presses and became unavailable for 3 min after the fifth lever press (i.e., depleted reward station; see Methods for task details). We recorded behavioral and neural data across 197 sessions (W: 101; Y: 96, see Supplementary Fig. 1 for the number of channels/ area). The average daily recording session was 97.8 minutes (SD ± 5.2 min).

We tracked the position of thirteen joints (key points) in our subjects with OpenMonkeyStudio[30]. We recorded neural activity using a data logger (SpikeGadgets, San Francisco, USA) attached to a multielectrode array (Gray Matter Research, Bozeman, USA) with 128 independently movable electrodes. We recorded both isolated neurons and local field potentials (LFPs) from eight areas: orbitofrontal (OFC), ventrolateral prefrontal (PFC), anterior cingulate (ACC), dorsolateral prefrontal (DLPFC) cortices, frontal eye fields (FEF), supplementary motor area (SMA), premotor cortex (PM), and the dorsal striatum (Fig. 1C). To quantify neural timescales, we focused on the local field potentials (see Fig. 1D and Methods) for two reasons: (1) spatial coverage, (2) their ability to be leveraged across long temporal windows allowing us to investigate the temporal dynamics of neural timescales throughout the recording session.

### Neural timescales are variable

We first examined session-wide neural timescales for individual recording sessions. For each recording site, we estimated neural timescales using a 10 s moving window with 5 s overlaps (see Fig. 2A and Methods). To identify the neural timescales characteristic of each brain area, we computed the median neural timescales collapsing across sessions and subjects. We demonstrated that neural timescales estimated from LFPs are ~10 times faster than those estimated from neuronal spiking data, i.e., ~10–50 ms, which is consistent with previous findings[16]. We further found that the magnitude of neural timescales decreased systematically across the duration of the recording session in all areas ($p < 0.05$, linear regression model, Fig. 2A).

We observed that task engagement (i.e., lever presses) also decreased across the recording session in both animals (Fig. 2B). Hence, we asked whether neural timescales and task engagement were related across the recording session. We hypothesized that there is a monotonic relationship between the two, with more task engagement being accompanied by slower dynamics. Specifically, we divided each session into ten equally sized (~10-min-long) segments. As a proxy for task engagement, we calculated the total number of lever presses in each segment. For every recording session, we computed the correlation between the magnitude of the neural timescales and our task

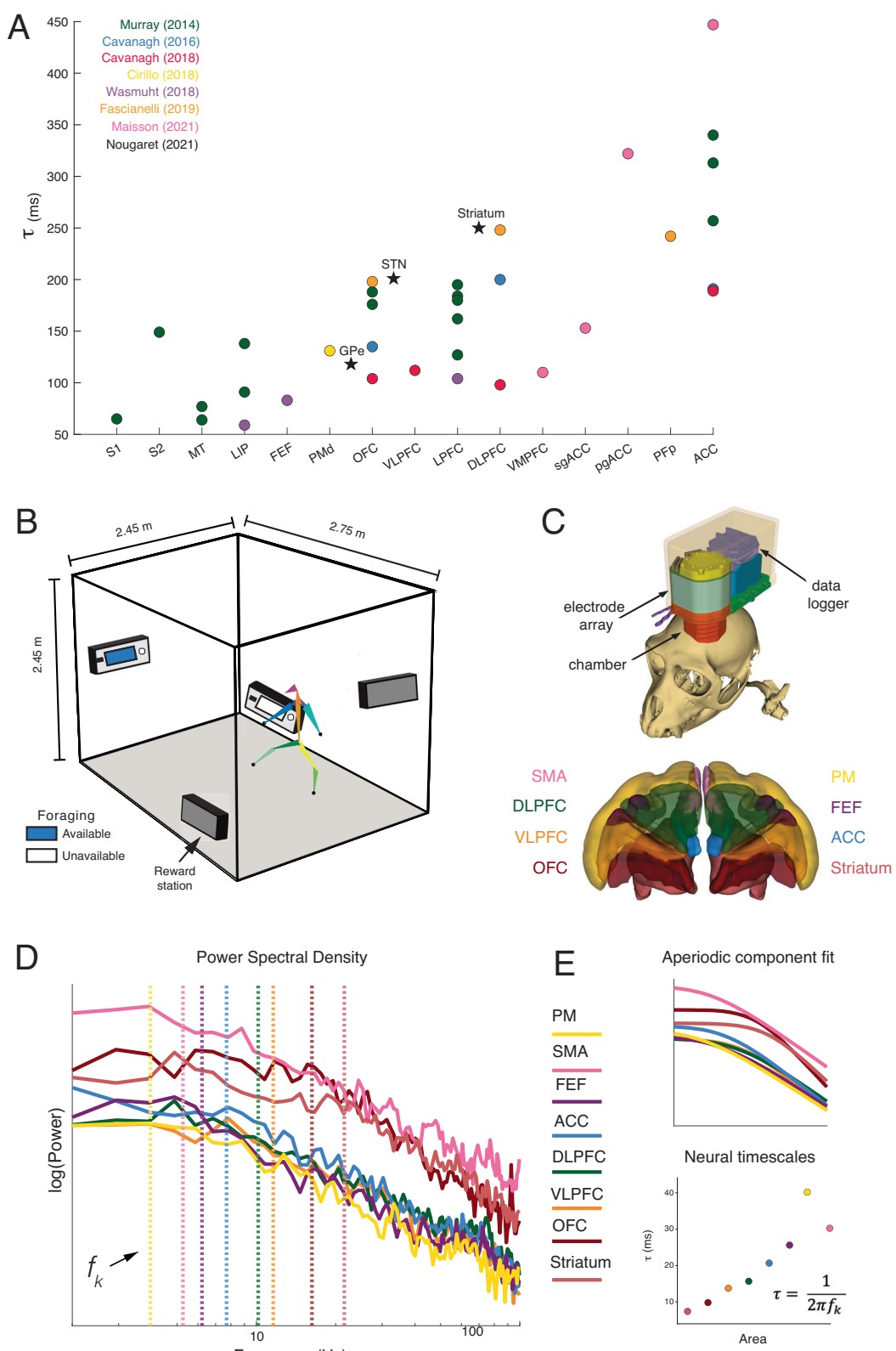

engagement index (see Supplementary Fig. 2H for the sample size of this analysis, as reflected by the number of recording sessions per area). For all areas, we observed a strong positive relationship between task engagement and the magnitude of the neural timescales, i.e., the median Pearson correlation coefficient across sessions ranged between 0.23 in the OFC and 0.65 in the VLPFC (Fig. 2C). To assess whether the median of the obtained distribution of Pearson

correlation coefficients was significantly larger than 0, we performed a one-sample Wilcoxon signed-rank for every area. In all areas, the median Pearson correlation coefficient was significantly larger than 0 ($p < 0.05$, with Bonferroni correction for eight comparisons).

Next, we asked whether neural timescales were related to movement, to account for the possibility that changes in neural timescales could be driven by increased motor activity, during intervals of high

**Fig. 1 | Overview of neural timescales estimation and experimental design.**
**A** Hierarchical organization of neural timescales at rest ($\tau$) estimated from neuronal spiking data. Neural timescales were estimated in 14 cortical and 3 subcortical areas). Traditionally, neural timescales were estimated in the pre-trial period of various tasks (i.e., chaired electrophysiology) by fitting an exponential decay function to the autocorrelation function (i.e., time-lagged correlation). Each circle represents the population-level $\tau$ for each cortical area and the stars represent population-level $\tau$ for each subcortical area reported in each study. **B** Depiction of the cage and foraging task. The subjects were allowed to freely explore and interact with reward stations in an open space - i.e., 2.45 × 2.45 × 2.75 m cage with barrels.

**C** Our recording system and recording sites in the striatum, OFC, VLPFC, DLPFC, ACC, FEF, PM, and SMA. **D** Local field potential (LFP) power spectral densities (PSDs) from example channels in Subject W. **E** Top: Aperiodic component fit for the example PSDs. We applied spectral parameterization to infer timescales from the PSDs (Donoghue et al., 2020; Gao et al., 2020). The periodic oscillatory peaks were discarded and the "knee frequency" ($f_k$ vertical dashed lines) was extracted from the fit of the aperiodic component. Bottom: Neural timescales ($\tau$) were inferred from $f_k$ via the embedded equation. OFC orbitofrontal cortex, VLPFC ventrolateral prefrontal cortex, DLPFC dorsolateral prefrontal cortex, ACC anterior cingulate cortex, FEF frontal eye fields, PM premotor cortex, SMA supplementary motor area.

displacement or motion in general. For every recording session, we computed the correlation between speed of displacement and neural timescales (see Supplementary Fig. 2L for the sample size of this analysis, as reflected by the number of recording sessions per area). We found that neural timescales were only weakly correlated with movement speed, i.e., the median Pearson correlation coefficient across sessions ranged from 0.05 in the PM cortex to 0.14 in the VLPFC (Fig. 2D). To assess whether the median of the obtained distribution of Pearson correlation coefficients was significantly larger than 0, we performed a one-sample Wilcoxon signed-rank for every area. In all areas, the median Pearson correlation coefficient was significantly larger than 0 ($p < 0.05$, with Bonferroni correction for eight comparisons). In combination, these results suggest that the gradual decrease in neural timescales throughout the recording session was primarily related to task engagement, an aggregate of behavioral state parameters in our task, and less so to movement parameters such as speed, a hypothesis we will further test next.

To test the unique relationship between speed, task engagement and neural timescales for the brain areas we recorded from, as well as to control for temporal autocorrelations, we next performed the following analysis. For each area, we randomly sampled without replacement $n$ (i.e., equivalent to the number of sessions) observations out of the total number of data points (note: the total number of data points per area can be calculated as the number of sessions × 10, i.e., number of time segments; see Supplementary Fig. 2J for the sample size of this analysis, as reflected by the number of recording sessions per area). For each subsample, we fit a linear regression model (see Supplementary Fig. 3 for the resulting distributions of standardized $\beta$ coefficients). We demonstrated that the effect of task engagement was gradually stronger in more dorsal areas—i.e., the regression coefficients were progressively larger ($p < 0.05$, pairwise two-sided independent sample $t$-test with Bonferroni correction for 28 comparisons; Fig. 2E) displaying the following ventro-dorsal ordering: OFC <Striatum <VLPFC -DLPFC-SMA < FEF < ACC < PM. Conversely, the effect of speed was gradually stronger in ventral areas—i.e., the regression coefficients were progressively larger ($p < 0.05$, pairwise two-sided independent sample $t$-test with Bonferroni correction for 28 comparisons; Fig. 2E) displaying the following ventro-dorsal ordering: OFC > VLPFC > Striatum - ACC > DLPFC > SMA - FEF > PM. Moreover, the effect of task engagement was larger than that of speed in more dorsal areas: SMA, PM, FEF, ACC, and DLPFC ($p < 0.05$, two-sided paired $t$-test with Bonferroni correction for eight comparisons). In contrast, there was no significant difference between the two predictors in VLPFC and the striatum, and the relationship was reversed in the OFC (p < 0.05, two-sided paired $t$-test with Bonferroni correction for eight comparisons).

Next, we demonstrated a monotonic relationship between brain areas and session-wide neural timescales (b = 2.29, 95% CI = [2.27 2.31]; monotonic Bayesian regression model) with the following hierarchical organization: OFC <Striatum <VLPFC < ACC < DLPFC < FEF < PM < SMA. To systematically assess the stability of this hierarchy across time, we additionally ranked the areas at each time point and compared their ordering to the session-wide hierarchy of neural timescales by using Spearman rank correlation (average Spearman rank

correlation coefficient 0.99, SD 0.01). This analysis further supported the Bayesian regression results, demonstrating a monotonic relationship between areas, with the relative position in the hierarchy being stable across time. In sum, we found a ventro-dorsal timescale hierarchy that overlapped but also deviated from previous findings in important ways. Briefly, we found that motor-related areas displayed the slowest timescales, followed by DLPFC and ACC, which displayed intermediate timescales, and finally, ventral areas displaying the fastest timescales. While the ventral areas displaying faster timescales than dorsal areas replicates previous work on monkey neuronal spiking timescales, motor-related areas are relatively higher in our hierarchy.

## The hierarchy of neural timescales at rest is dependent on behavioral demands

To test if baseline neural timescales, i.e., at rest, are also shaped by this context, we estimated neural timescales during task-free periods of time[10]. We did not impose any task demands on our animals, and hence, there was no predefined resting-baseline before a trial (i.e., the intertrial interval was self-imposed). As a result, there was a wide repertoire of behaviors in the moments before a lever press (e.g., sitting, walking, etc.). To capture moments when the animal was at "rest", we operationalized task-free trials as periods of 5 s (or longer) during which displacement of the 3D center-of-mass was less than 40 cm, excluding task engagement (see Fig. 3A and Methods). The average task-free period length was 6.5 s (SD = 0.4 s).

Using this approach, we showed that the hierarchy of task-free neural timescales was consistent with our session-wide results (see Supplementary Fig. 2K for the sample size of this analysis, as reflected by the number of recording sessions per area). We found the following hierarchy when the animal was at "rest": OFC <Striatum - VLPFC < DLPFC < ACC < FEF < PM-SMA (see Fig. 3B). The magnitude of the neural timescales differed significantly between areas, except for the following pairs: PM-SMA, ACC-FEF, and VLPFC-striatum ($p < 0.05$, pairwise two-sided Mann–Whitney $U$-test computed across sessions with Bonferroni correction for 28 comparisons). In sum, we show that the baseline itself is not intrinsic but rather a reflection of the contextual cognitive and perceptual demands imposed on the brain.

## Neural timescales changes are dependent on the behavioral context

Given the variability in neural timescales described above, we hypothesized that changes in neural timescales could depend on the behavioral context. In our experiment, the primary behavioral demands on the monkeys resulted from the pattern of engagement with the reward stations, i.e., the decisions to engage or disengage with a particular reward station location. We therefore divided events with lever presses into three categories: (1) first press on a feeder, (2) intermediate presses, and (3) final lever presses. We hypothesized that these three types of lever presses could be associated with different neural timescales signatures because, even though they have identical actions, they have inherently different cognitive meanings and behavioral contextual demands. The first lever press reflects the decision to forage at a given reward station, while the final lever press ends that goal sequence and requires an animal to decide on what to do next.

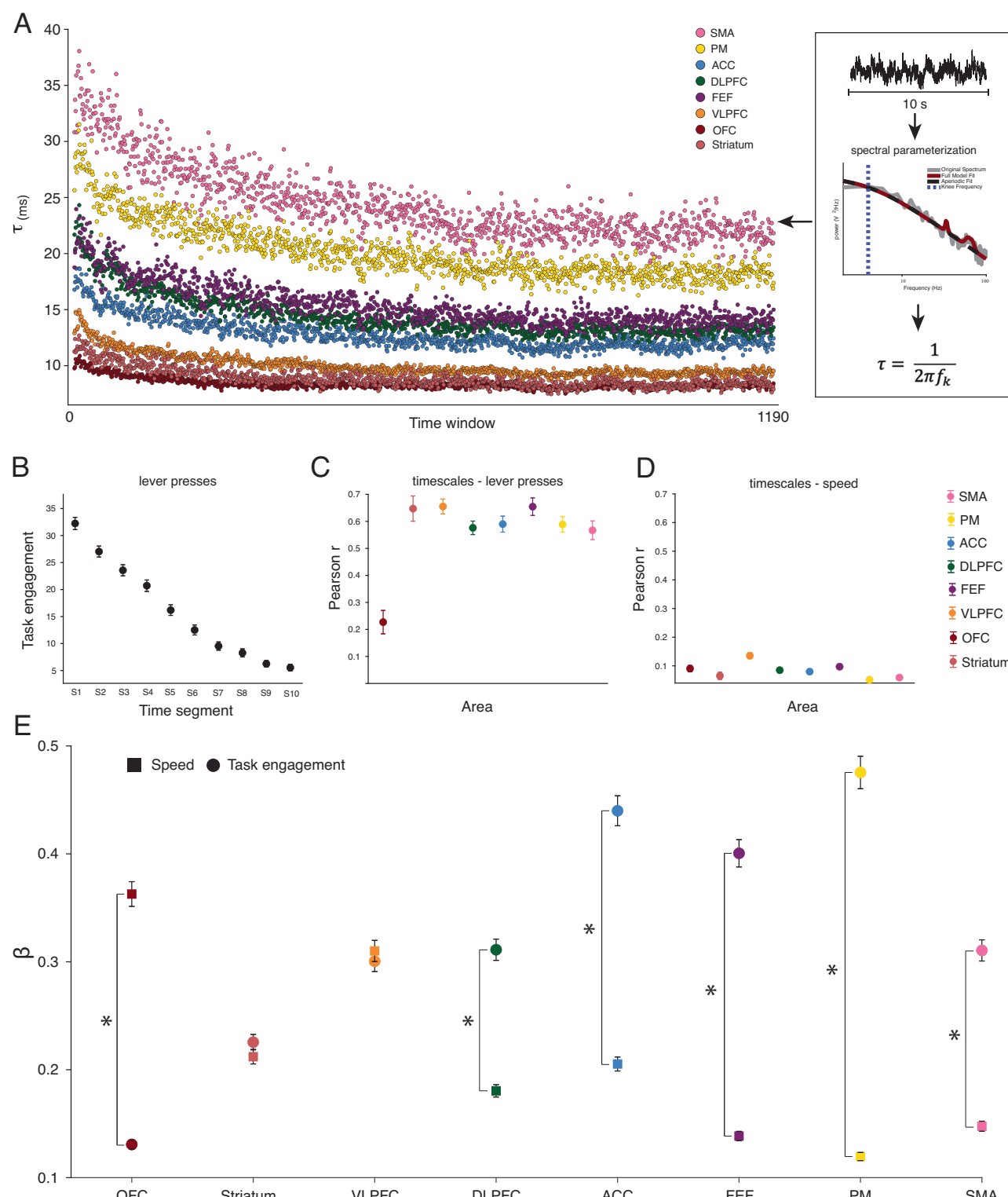

Intermediate lever presses are more heterogeneous in terms of their position in the goal sequence and were therefore not considered for this analysis.

Because we were particularly interested in changes before and after the lever presses, we time-locked the estimation of the neural timescales to the event to examine how the event itself affects these dynamics (Fig. 3A). We excluded the window centered on the event since it would include the dynamics of both before and after the decision. We found an expansion in the magnitude of timescales during task engagement without an associated change in the relative

position of an area in the overall hierarchy−i.e., all areas showed significant $\tau$ increase compared to baseline (defined as the median task-free timescales at rest) at any time point before and after the event ($p < 0.05$, one-sided Wilcoxon signed-rank test with Bonferroni correction for 160 comparisons).

Interestingly, a clear and unique temporal dynamic profile separated the two categories of lever presses. For the first lever presses, we found that neural timescales gradually increased in the seconds leading up to the interaction with the reward station and sharply decreased in the seconds after (Fig. 3C). To assess the significance of this linear

**Fig. 2 | Session-wide timescale dynamics and correspondence to behavioral variables. A** Right: Neural timescales ($\tau$) estimated for individual recording sessions from Power Spectral Densities (PSDs) Left: Neural timescales across time, collapsed across recording sessions and subjects. The hierarchical ordering of the areas is conserved across the duration of the recording session. Y-axis: median neural timescales. X-axis: time widows. **B** Average number of lever presses, a putative index of task engagement, decreases over time. Y-axis: average number of lever presses across sessions in each 10 min time segment ($N = 197$). Bars indicate the standard error of the mean across recording sessions. **C** Task engagement was correlated with neural timescales in all areas. Y-axis: the median correlation coefficient between the number of lever presses and neural timescales across time (see supplementary Fig. 2H for sample size). Neural timescales and task engagement were correlated for individual recording sessions using the 10 min time segments. Circles: brain areas, median ± s.e.m across recording sessions. **D** Speed of movement and neural timescales were weakly correlated. Neural timescales and speed

were correlated for individual recording sessions. Speed of movement was downsampled by using a moving average 10 s window (with 5 s overlap) to match the neural timescales. Y-axis: the median correlation coefficient between speed and neural timescales across time (see supplementary Fig. 2I for sample size). Circles: median ± s.e.m across sessions. **E** The effect of task engagement is gradually stronger and more separable from the effect of speed as we move from ventral to dorsal areas. Y-axis: average regression coefficients across iterations ($N = 1000$). Squares: mean $\beta$ (speed of movement) ± s.e.m. Circles: mean $\beta$ (task engagement) ± s.e.m. Asterisk: statistical significance (two-sided paired t-test) between predictors within an area at $p < 0.05$, with Bonferroni correction ($N = 8$). OFC orbitofrontal cortex, VLPFC ventrolateral prefrontal cortex, DLPFC dorsolateral prefrontal cortex, ACC anterior cingulate cortex, FEF frontal eye fields, PM premotor cortex, SMA supplementary motor area. Source data are provided as a Source Data file.

increase for individual areas, we conducted a linear regression model for each recording session (see Supplementary Fig. 2L for the number of recording sessions per area for the first and final lever presses), with time as the predictor and neural timescales as the dependent variable (see Supplementary Fig. 4A, B for the resulting distribution of standardized regression coefficients for the first and final lever presses). Before the first lever press, for all areas across recording sessions, the resulting regression coefficients were significantly larger than 0 ($p < 0.05$, one-sided Wilcoxon signed-rank test with Bonferroni correction for 32 comparisons). After the event, in some areas such as SMA, FEF, ACC, VLPFC, and striatum neural timescales started increasing again, while for the others, there was no significant trend across time ($p < 0.05$, one-sided Wilcoxon signed-rank test with Bonferroni correction for 32 comparisons). For the final lever presses, neural timescales gradually increased in the seconds leading up to and continued to increase after the interaction with the reward station (Fig. 3D). Similar to the first lever presses, we found that for all areas across events, the resulting regression coefficients were significantly larger than 0 before the event (p < 0.05, one-sided Wilcoxon signed-rank test with Bonferroni correction for 32 comparisons). In contrast, in all areas we found that regression coefficients were significantly smaller than 0 after the event ($p < 0.05$, one-sided Wilcoxon signed-rank test with Bonferroni correction for 32 comparisons).

So far, we have therefore demonstrated a nested correspondence between neural timescales and the temporal scales over which behavior unfolds. At long (session-wide) temporal scales, neural timescales corresponded with overall task engagement, as shown above, while at short temporal scales, neural timescales exhibited variability corresponding to ongoing behavioral demands from our task.

## The temporal adaptation of neural timescales varies by area
In the previous section, we demonstrated a correspondence between neural timescales and behavioral contextual demands. Although we showed that the hierarchy expands in the seconds leading up to the lever presses, we did not assess whether this magnitude change differed by area. Here, we hypothesized that this magnitude expansion of neural timescales might indeed be differentially modulated per area. To that end, we examined the pairwise differences between areas (two-sided Mann–Whitney $U$-test with Bonferroni correction for 280 comparisons per event), with respect to their change from resting-baseline for each time point before and after the first and final lever presses (Fig. 4; see Supplementary Fig. 2N, O for the sample size of these analyses, as reflected by the number of recording sessions per area). Note that we chose this statistical approach for the sake of robustness and to avoid overfitting. We normalized the change in neural timescales per area by subtracting the respective area-specific resting-baseline ($\Delta\tau$) for each time point. We found that this adaptation differed in magnitude by area, such that while the change was undifferentiated several time points before

the event, a differentiated hierarchy emerged just before the events of interest.

Notably, we found that a ventral to dorsal grouping of the areas emerged before each event: (1) the striatum and OFC displayed the smallest change in timescale magnitude; (2) the VLPFC, DLPFC, and ACC displayed an intermediate level of magnitude change that was significantly higher than in the striatum and OFC; (3) and the FEF, PM and SMA displayed the highest level of magnitude change—hence, these areas were placed at the top of the hierarchy of change (see table inserts in Fig. 4 for statistics). We found this grouping both before the first and final lever presses. However, each category of events exhibited unique temporal profiles after the event. Notably, for the first lever presses, the areas clustered right before the event and became undifferentiated immediately after (see table inserts in Fig. 4 for statistics). For the final lever presses, similar clusters to those observed for the first lever presses emerged before the event, but they persisted after the event. Additionally, in support of the idea that there are area-specific modulations, see Supplementary Fig. 5 that depicts the change as a percentage of baseline. In summary, while task engagement generally expanded the magnitude of neural timescales, this expansion was differentially modulated by the task demands—i.e., it depended on the type of event and the brain area. To quantify these two types of observed changes (i.e., global, and area-specific changes), for each time point, we calculated the effect size of each Mann–Whitney $U$-test by estimating the rank-biserial correlation coefficient[34]. Next, for each time point, we averaged the absolute value of the effect sizes across the Mann–Whitney $U$-tests (i.e., associated with each time point's 28 pairwise comparisons). For each time point, the average rank-biserial correlation coefficient was used as an index of how differentiated the areas were in their change from baseline. This index ranges from 0, when all areas display the exact same change from baseline, to 1, which would indicate complete independence in their change from baseline (see figure below). Supporting our previous conclusion, the differentiation between areas increased in the moments leading up to the lever presses. However, the two types of events (first and final lever presses) diverge after the event. In particular, the first lever presses display a drop, while the final lever presses display a continuation of this differentiation between areas. We conclude that there are both global changes, potentially driven by similar mechanisms across areas, but also more targeted effects that differentiate the areas in their response to foraging events.

## The dynamics of neural timescales reflect key foraging events
Finally, we investigated whether timescale adaptations differed between events with lever presses that have the same position in the sequence but are followed by a different outcome. This way, we aimed to further dissect behavior with respect to its cognitive meaning and behavioral demands. To operationalize this, we looked at the fourth lever press, which can indicate different behavioral motifs depending

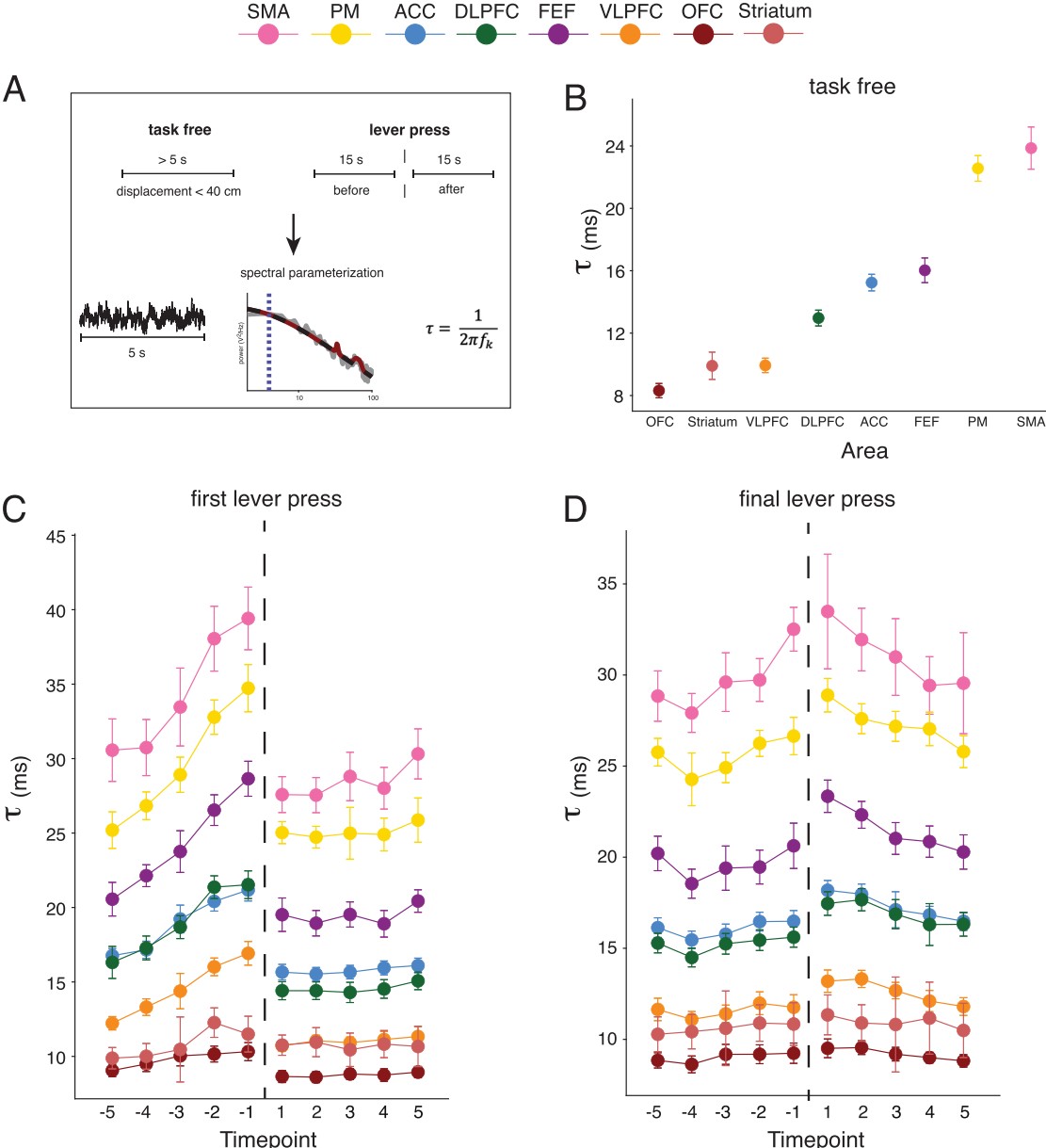

**Fig. 3 | Neural timescales for events with different behavioral contexts. A** Neural timescales ($\tau$) estimation for task-free segments and lever presses. **B** Task-free neural timescales are hierarchically organized. Circles: median across sessions ± s.e.m. (see supplementary Fig. 2K for sample size). **C** Neural timescales surrounding the first lever press. Vertical dotted line: time of lever press. Circles: median across sessions ± s.e.m. (see supplementary Fig. 2L for sample size). **D** Neural timescales

surrounding the final lever press. Vertical dotted line: time of lever press. Circles: median across sessions ± s.e.m. (see supplementary Fig. 2M for sample size). OFC orbitofrontal cortex, VLPFC ventrolateral prefrontal cortex, DLPFC dorsolateral prefrontal cortex, ACC anterior cingulate cortex, FEF frontal eye fields, PM premotor cortex, SMA supplementary motor area. Source data are provided as a Source Data file.

on the sequence of foraging bouts. For example, a monkey can leave a feeder after four lever presses without performing the fifth press to time out the system to go to the next feeder or engage in a different behavior. From the perspective of an action, there is no difference between timing the system out versus choosing to disengage early. However, we hypothesized that neural timescales could exhibit differentiable temporal profiles also to these more intricate behavioral sequence differences. Practically, we compared neural timescales on the fourth lever press when the monkey decided to leave versus when they decided to stay for a fifth lever press (see Fig. 5A). Hence, we hypothesized that the before-after dynamics around "leave" would be similar to that of the final lever press. In contrast, we hypothesized that there would be no significant before-after differences for the "stay" lever presses. For each event and area, we therefore computed the

pairwise differences between before and after changes in the magnitude of neural timescales from resting-baseline (see Supplementary Fig. 2N–P for the sample size of this analysis, as reflected by the number of recording sessions per area). In other words, this analysis compared the time point preceding and following a lever press event in terms of their magnitude change from resting-baseline. We found that for all areas, the before-after change from resting-baseline to the first lever press was characterized by a significant attenuation, with slower neural timescales before and faster neural timescales after the event ($p < 0.05$, one-sided Wilcoxon signed-rank test with Bonferroni correction for 8 comparisons; Fig. 5B). The final lever press displayed the opposite pattern, with a significant increase in the change from resting-baseline, and ultimately slower neural timescales after the event ($p < 0.05$, one-sided Wilcoxon signed-rank test with Bonferroni

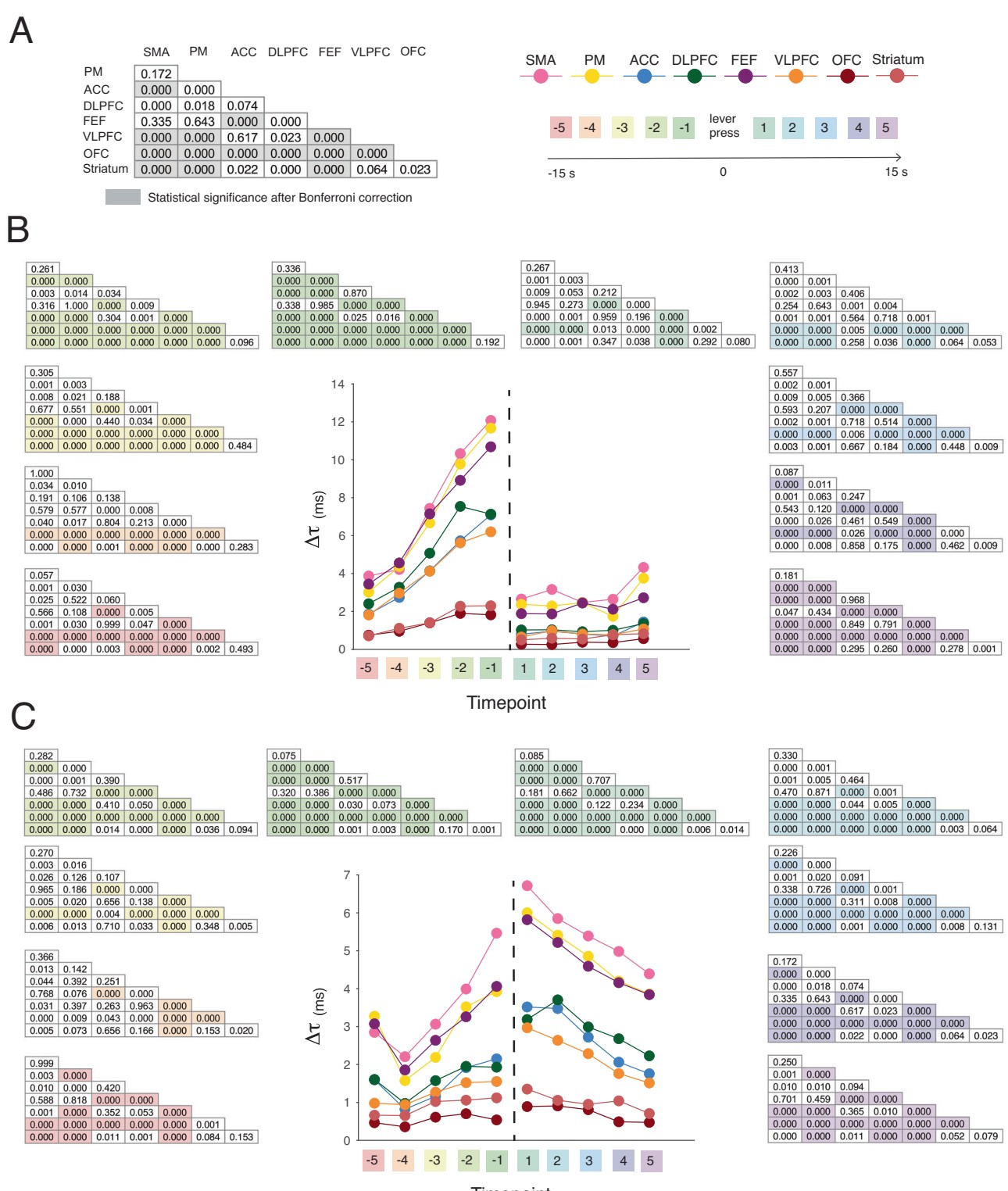

**Fig. 4 | The area-specific adaptation of neural timescales. A** We examined pairwise differences between areas' change from the area-specific resting-baseline ($\Delta\tau$) for each time point before and after the event of interest. Left: example statistical table with $p$ values for the two-sided Mann–Whitney $U$-test with the shaded areas representing statistical significance at $p < 0.05$ after Bonferroni correction. Right: the color scheme of the individual areas and the color scheme of different time points (Note: used for indicating the associated statistical table). **B** The change from the area-specific resting-baseline for the time points before and after the first lever press. The tables depict the $p$ value associated with each pairwise comparison for the five time points before and after the lever press. The shading represents statistical significance at $p < 0.05$ with Bonferroni correction

($N = 280$). Circles: median across sessions. (see supplementary Fig. 2N for sample size). **C** The change from area-specific resting-baseline for the time points before and after the final lever press. The tables depict the $p$ value associated with each pairwise comparison for the five time points before and after the lever press. The shading represents statistical significance at $p < 0.05$ with Bonferroni correction ($N = 280$). Circles: median across sessions. (see supplementary Fig. 2O for sample size). OFC orbitofrontal cortex, VLPFC ventrolateral prefrontal cortex, DLPFC dorsolateral prefrontal cortex, ACC anterior cingulate cortex, FEF frontal eye fields, PM premotor cortex, SMA supplementary motor area. Source data are provided as a Source Data file.

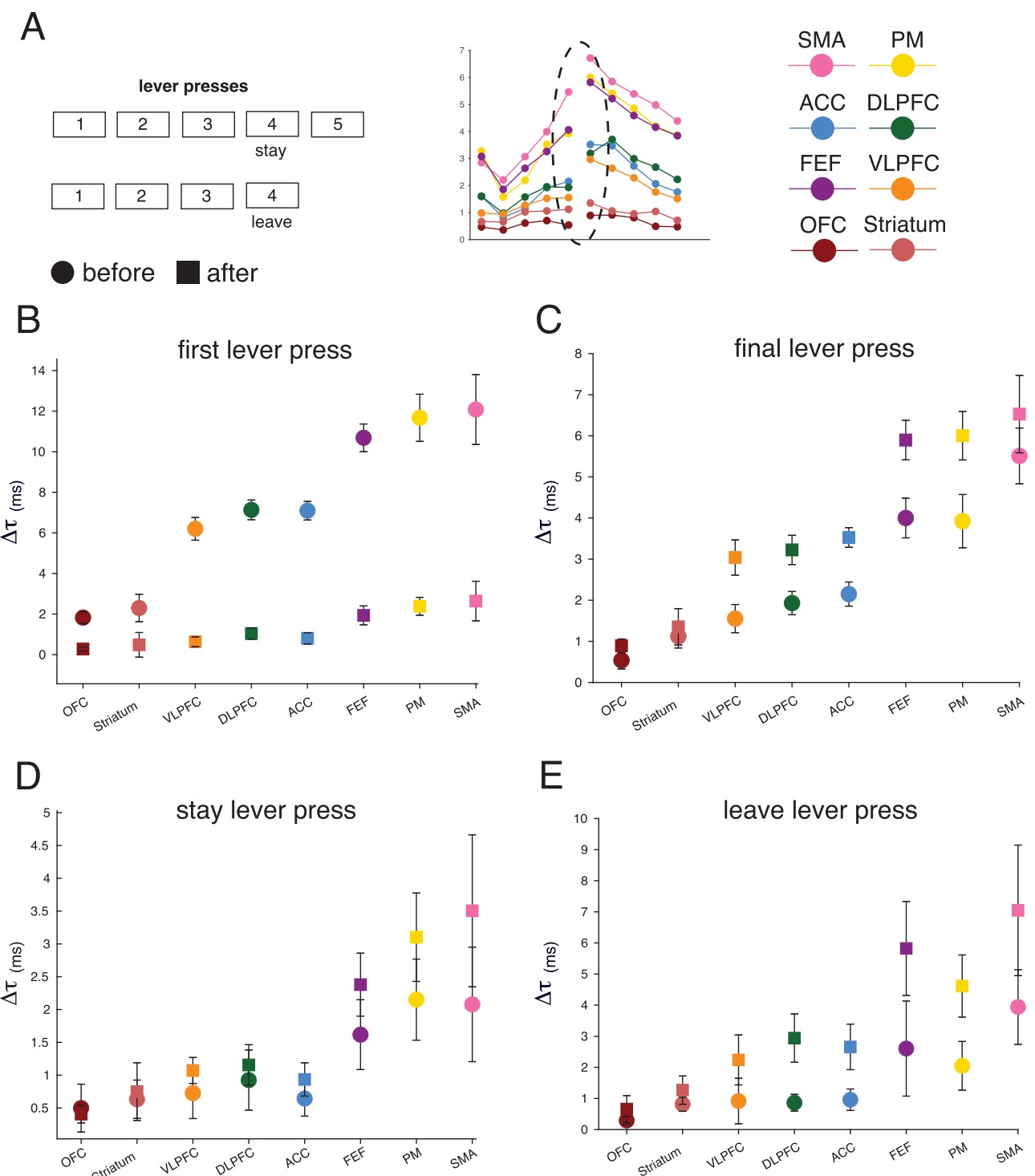

correction for eight comparisons; Fig. 5C). Note that for visualization purposes, six outliers for SMA and one outlier for FEF were excluded. Although outliers do not affect tests such as the Wilcoxon signed-rank test, they bias the standard error, which is then not representative of the spread of the area's data. As hypothesized, "stay" lever presses did not elicit significant changes in the before-after dynamics of any area— i.e., the event was not accompanied by a unique neural timescales signature ($p < 0.05$, two-sided Wilcoxon signed-rank test with Bonferroni correction for eight comparisons; Fig. 5D). In contrast, leave lever presses were accompanied by a significant before-after expansion of the magnitude of neural timescales which mimicked that of the final lever presses ($p < 0.05$, one-sided Wilcoxon signed-rank test with

Bonferroni correction for eight comparisons; Fig. 5E). Overall, we demonstrate that the event-related changes in neural timescales are dependent on their abstract meaning in the context of the foraging task (see Fig. 6C for an overview).

## Discussion

The brain is characterized by a hierarchical gradient of neural timescales in both human and nonhuman primates[9,11,14], which is assumed to arise from macroscale and microcircuit anatomical and functional connectivity, as well as variation in cytoarchitecture[10,20]. The question arises as to how these temporal properties of the brain give rise to adaptive behavior that requires flexible adjustment of temporal coding

**Fig. 5 | Neural timescales dynamics reflect fine-grained abstract meaning.**
**A** Neural timescales ($\tau$) estimation for three categories of lever presses: first lever press, intermediate ("stay") lever press, and final lever press. **B** The change in neural timescales from resting-baseline before and after the first lever presses. Y-axis: change in neural timescales from resting-baseline ($\Delta\tau$). The change in neural timescales significantly decreased from before to after the lever press in all areas ($p < 0.05$, one-sided Wilcoxon signed-rank test with Bonferroni correction for eight multiple comparisons). Squares: median across sessions $\pm$ s.e.m for after the event. Circles: median across sessions $\pm$ s.e.m for before the event (see supplementary Fig. 2N for sample size). **C** The change in neural timescales from resting-baseline before and after the final lever presses. Y-axis: change in neural timescales from resting-baseline ($\Delta\tau$). The change in neural timescales significantly increased from before to after the lever press in all areas ($p < 0.05$, one-sided Wilcoxon signed-rank test with Bonferroni correction for eight multiple comparisons). Squares: median across sessions $\pm$ s.e.m for after the event. Circles: median across sessions $\pm$ s.e.m for before the event (see supplementary Fig. 2O for sample size). **D** The change in neural timescales from resting-baseline before and after the stay lever presses. The change in neural timescales did not significantly change from before to after the lever press in any area ($p < 0.05$, two-sided Wilcoxon signed-rank test with Bonferroni correction for 8 multiple comparisons). Squares: median across sessions $\pm$ s.e.m for after the event. Circles: median across sessions $\pm$ s.e.m for before the event (see supplementary Fig. 2P for sample size). **E** The change in neural timescales from resting-baseline before and after the leave lever presses. Y-axis: change in neural timescales from resting-baseline ($\Delta\tau$). The change in neural timescales significantly increased from before to after the lever press in all areas ($p < 0.05$, one-sided Wilcoxon signed-rank test with Bonferroni correction for eight multiple comparisons). Squares: median across sessions $\pm$ s.e.m for after the event. Circles: median across sessions $\pm$ s.e.m for before the event (see supplementary Fig. 2P for sample size). OFC orbitofrontal cortex, VLPFC ventrolateral prefrontal cortex, DLPFC dorsolateral prefrontal cortex, ACC anterior cingulate cortex, FEF frontal eye fields, PM premotor cortex, SMA supplementary motor area. Source data are provided as a Source Data file.

and integration demands. Here we found a ventro-dorsal hierarchy of neural timescales that is influenced by this foraging environment. Importantly, we showed that this hierarchy is preserved even in the context of flexible task demands. However, the magnitude of neural timescales dynamically expanded depending on overall task engagement over long temporal scales, but also varied with the cognitive demands of our task over shorter temporal scales. Notably, we observed systematic changes in the magnitude of neural timescales that span the duration of a recording session in both animals and during every recording session, both the magnitude of neural timescales and task engagement gradually and reliably decreased over time. Importantly, these results were not driven by variability in motor-related activity. Within these global session-wide changes, we found variability in the magnitude of neural timescales that is associated with the abstract cognitive meaning of the different foraging events. Hence, the neural timescales change patterns differentiated between fine-grained behavioral states. Our results are evidence that the multitude of external temporal scales over which behavior unfolds is mirrored by changes in neural timescales that occur at multiple scales, with local foraging-related changes nested within general engagement-related changes that span longer temporal scales.

Contrary to previous work on timescales, which was usually done in the context of memory-related or value-encoding tasks[8,35], we examined self-paced unconstrained behavior that was not focused on an isolated cognitive component. We consistently found a stable ventral to the dorsal hierarchy of neural timescales that extended from OFC to motor-related areas. This was the case across all our analyses, i.e., for neural timescales estimated during unconstrained movement, around foraging events, and even in the absence of task engagement. Our observed hierarchy partially overlapped but also deviated from previous work (see Fig. 6A for an overview of previous findings). The first major discrepancy was that the ACC, usually displaying the slowest timescales[10,15], was not at the top of our observed hierarchy, but rather exhibited intermediate timescales. Moreover, motor-related areas, which previously displayed faster timescales relative to prefrontal areas in nonhuman primates[23], exhibited the slowest temporal dynamics in our foraging environment. This was actually in full agreement with results from a rodent paradigm that allows for whole-body movement[36]. We believe these results originate from our experimental paradigm that allowed for unconstrained movement in a large open space. This contrasted with studies using chaired paradigms that require head fixation and restrained body movement. In line with our initial hypothesis, our results supported the idea that environmental demands shape the observed hierarchy of neural timescales. This is not surprising since behaviorally relevant neural timescales imply a certain amount of dynamic range. Deviations from the commonly reported hierarchical organization of neural timescales estimated from monkey neuronal spiking have been previously

described in a study using human LFP data[16]. In particular, the OFC displayed slower timescales than other PFC structures, which is the opposite of what had been previously reported (see Fig. 6A for comparisons). It is important to note that our results are not incompatible with the previous literature on hierarchies of neural timescales at rest but are rather complementary by investigating neural timescales in a new context, that of unconstrained behavior. While these differences in the relative position of an area within the hierarchy could be due to using different modalities, previous work demonstrating a strong correspondence between signal modalities[16] suggests they are the result of contextual adaptation of the observed hierarchy of neural timescales. Our results further expanded previous work as we simultaneously estimated neural timescales in the dorsal striatum and prefrontal cortical structures, giving us the opportunity to directly place the dorsal striatum in the context of the broader extensively studied cortical hierarchy. The striatal neural timescales reported here place the striatum on a comparable level to ventral prefrontal areas, which is in agreement with previous reports of neural timescales in this subcortical structure[37]. Overall, we confirmed our hypothesis that the observed hierarchy of neural timescales emerges from the particular input-output demands imposed on the brain within the bounds of what the anatomical network permits.

Matching previous work on neural timescales at rest, i.e., estimated during the baseline or pre-trial period (for an overview, see Figs. 1A, 6A), we estimated neural timescales during immobility periods. Since the animals were free to decide when or even if, to engage in the task, the current paradigm did not have a traditional pre-trial period comparable to previous work—i.e., the repertoire of behaviors before engaging with a reward station is highly heterogeneous. Immobility periods, when the animal is disengaged from the task, were the closest match to capturing the animal at rest. It is important to note that rest or baseline is generally difficult to define—the traditional fixation period used to define neural timescales at rest is assumed to include little task-relevant signals. However, a lack of outward behavior does not imply a lack of cognitive processing. Indeed, previous findings on the relationship between neural timescales at rest, estimated during the intertrial period, and strength of neural encoding during a task did not always replicate[21]. This might be a result of task-relevant cortical activity emerging, or remaining during the fixation period[21]. To estimate neural timescales at "rest", work in anesthetized subjects or during sleep might be a viable alternative[9,38,39], although these approaches come with their own disadvantages and confounds. We have previously shown that neural timescales estimated from fMRI data in anesthetized nonhuman primates replicated hierarchies derived from neuronal spiking data, although they did not perfectly match[9]. Amongst many potential reasons for the observed deviations, one could be that anesthesia provides a special controlled state. Alternatively, electrophysiological recordings during dedicated rest

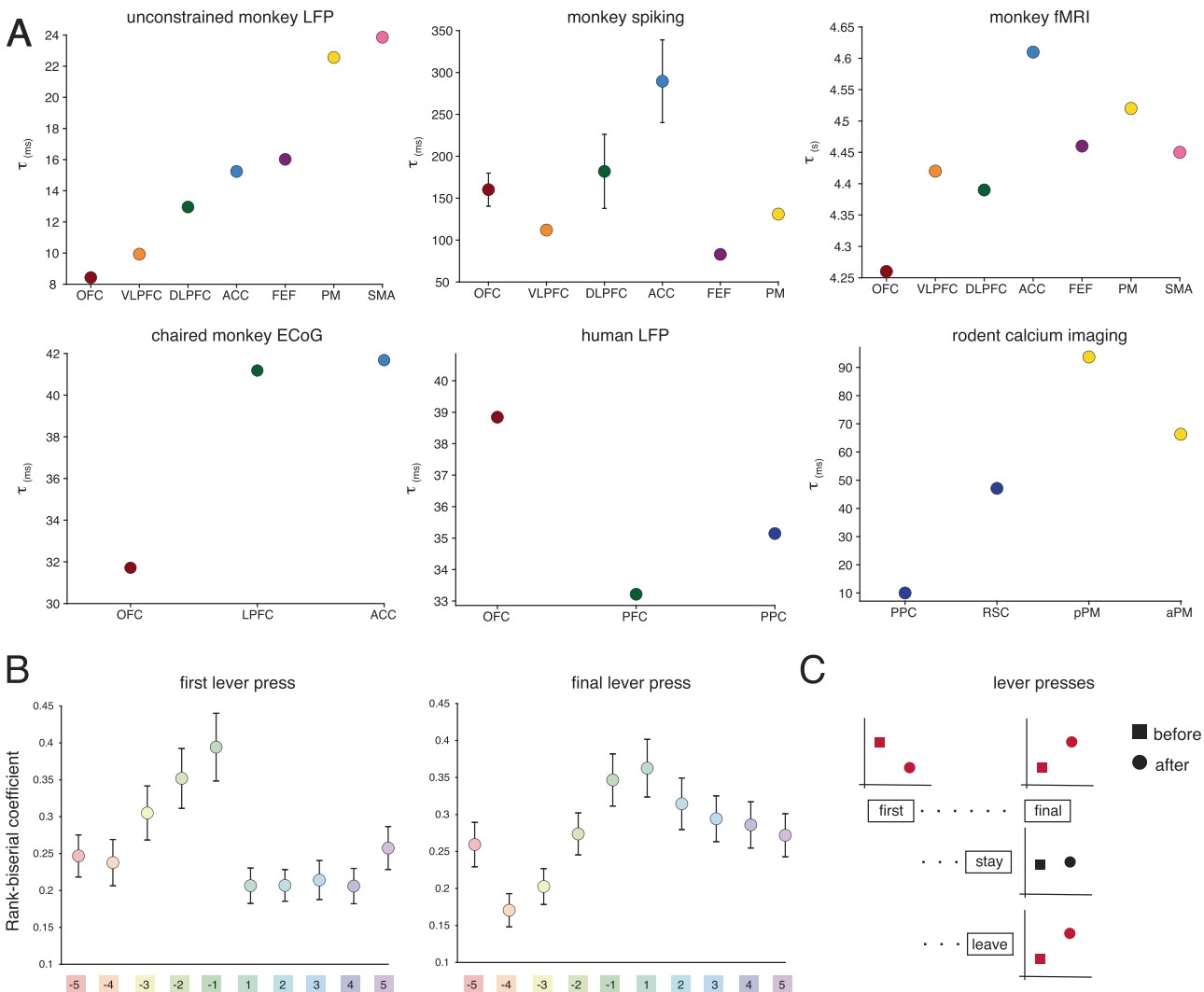

**Fig. 6 | Overview of the dynamics of neural timescales. A** The hierarchical organization of cortical neural timescales across species and modalities. Unconstrained monkey LFP: the task-free hierarchy of neural timescales (see Fig. 3A). Chaired monkey ECOG: the hierarchy of neural timescales estimated from monkey ECoG data as reported in Gao et al., 2020. Monkey spiking: the hierarchy of neural timescales estimated from neuronal spiking across studies (see Fig. 1A). The error bars represent ±s.e.m. across studies. Monkey fMRI: the hierarchy of neural timescales estimated from monkey fMRI data as reported in Manea et al., 2022. Human LFP: the hierarchy of neural timescales estimated from human LFP data as reported in Gao et al., 2020. Rodent calcium imaging: the hierarchy of neural timescales estimated from rodent calcium imaging data as reported in Pinto et al., 2022. **B** Global and event-specific changes in neural timescales. We estimated the effect size of the separation between areas' change from baseline by calculating the rank-biserial correlation for each statistical test in Fig. 4. We use the average effect size at each time point as an index of the differentiation between areas, with low values reflecting a global change in neural timescales (i.e., most areas displayed the same level of change), and high values reflecting area-specific changes (i.e., most areas were statistically different from each other). Y-axis: the average rank-biserial correlation ± s.e.m. across multiple comparisons (i.e., 28 comparisons per time point). X-axis: time point before and after the lever presses, with the colors indicating the associated statistical table in Fig. 4. Circles: mean across statistical tests ± s.e.m ($N$ = 28). **C** Overview of the before-after differences across events with seemingly similar motor and reward properties, but different abstract meanings. The figure depicts the general effects as observed across areas. For the first lever press, neural timescales decrease after the event. For the final lever presses, neural timescales increase after the event. For the leave lever presses, we observe the same effect. For stay lever presses, there are no significant changes in the before-after change in neural timescales. Circles: changes after the event. Squares: changes before the event. Red: statistically significant before-after changes across all areas. OFC orbitofrontal cortex, VLPFC ventrolateral prefrontal cortex, DLPFC dorsolateral prefrontal cortex, ACC anterior cingulate cortex (ACC), FEF frontal eye fields FEF, PM premotor cortex, SMA supplementary motor area. Source data are provided as a Source Data file.

periods, in the absence of any task similar to the human resting-state fMRI literature, could shed light on this issue. We speculate that while anatomy constrains the space of possible neural timescales, contextual behavioral demands modulate the observed hierarchy even at rest. Interestingly, the hierarchy of neural timescales we observed mirrored functional hierarchies that we found in this dataset with respect to action[32] and spatial navigation encoding[31]. In these studies, encoding of spatial navigation and action-related variables was progressively stronger from ventral to dorsal areas[31,32], and hence potentially facilitated by longer temporal processing or integration windows as

reflected by slower neural timescales. Our current timescale findings and the previous task variable encoding findings, match our previous work demonstrating that the hierarchy of neural timescales at rest in the medial PFC closely follows a ventro-dorsal functional hierarchy of the decodability of choice-relevant task variables—i.e., encoding of task-related variables is stronger in areas with longer neural timescales[15].

We found that neural timescales expanded during task engagement, in agreement with previous studies[16,18]. Interestingly, we found a monotonic relationship between the extent of task engagement and

the overall magnitude of neural timescales over a long temporal scale which spans the recording session. More task engagement was accompanied by an expanded hierarchy of overall slower neural timescales throughout the recording session. Within this session-wide change with behavioral engagement, we demonstrated local event-related changes in the magnitude of neural timescales. Although the action of pressing a lever was similar irrespective of its location in the broader foraging context, unique temporal dynamics of neural timescales were associated with differences in cognitive meaning. We speculate that this is a result of differences in the underlying computations associated with these lever presses. Neural timescales thus seemed to track the temporal persistence of information relevant during the ongoing decision process in a behaviorally relevant manner. For example, while the first lever press reflects the decision to forage at a particular reward station, the action per se can be seen as the end goal. Interestingly, this is accompanied by a significant drop in the magnitude of neural timescales. We speculate that accomplishing this goal could act as a stop signal for integration. In contrast, the last lever press seemed to reflect ongoing integration related to the animal needing to decide what to do next, with neural timescales continuing to expand after the event. Nevertheless, the specific meaning of these time-locked local changes in the magnitude of neural timescales remains an outstanding question. Although ecologically valid, unconstrained behavior automatically introduces variability that cannot be controlled for, nor is easily modeled. Our results, therefore, could be extended by directly manipulating the information integration across multiple temporal scales, while introducing clear task endpoints.

Despite widespread modulation by general task demands, we also found that the temporal dynamics of neural timescales showed time-locked changes to certain foraging events that differed by area (see Fig. 6B). We found a significant differentiation across areas in the time-locked expansion of the hierarchy linked to certain foraging events. Notably, however, the ventral to dorsal grouping of areas was preserved during this expansion—i.e., the dorsal striatum and OFC exhibited the lowest level of change from resting-baseline; VLPFC, DLPFC, and ACC exhibited intermediate levels; and FEF, PM, and SMA were at the top of the hierarchy, displaying the highest level of change from resting-baseline. Importantly, although this ventro-dorsal hierarchy of change was present for all lever presses, its emergence and persistence depended on the cognitive meaning mentioned above. Changes in the real world happen on different timescales, and these results suggest that individual brain areas may contribute distinctly to behavioral adjustments on different temporal scales. It is not trivial to quantify unconstrained behavior, and it is even more challenging to infer the cognitive state of the animal in this setting. As a result, it remains an open question as to what particular task variables are associated with the observed variability.

Here we use LFP rather than single-unit activity to infer timescales for two reasons. First, LFP activity, in our dataset, offered much broader spatial and temporal coverage and provided us with the ability to record a large number of areas simultaneously. That is, we were able to leverage the nature of this signal to infer neural timescales in all areas across time throughout the recording session. Second, the neuronal firing rates in this dataset are sparse, and hence, it would have been difficult, if not impossible, to calculate the autocorrelation function across time for many cells. It is possible that the results reported here are unique to LFP activity and do not fully translate to single-unit activity. Nevertheless, it is important to note that the LFP signal itself is unique in that it is a continuous signal that can be leveraged across long temporal windows allowing us to probe the dynamics of neural timescales. An equivalent analysis could not be easily accomplished with single-unit activity. Our main goal was to use whatever brain signal allows us to investigate the temporal dynamics of neural timescales in this unconstrained foraging context from a population perspective. Notably, even timescales estimated from

neuronal spiking have been generally described and reported as a population-level statistic due to their high heterogeneity exhibited at the single-neuron level (see ref. 10). That being said, although LFP and single-unit activity are fundamentally different signals, Gao et al., 2020 showed that macaque ECoG timescales track previously published spiking timescales (Murray et al., 2014). Moreover, we have previously shown that neural timescales estimated from neuronal spiking and fMRI, a modality closely related to LFP[40], display a very similar hierarchical organization[9]. There is also evidence that the unimodal areas to association areas axis of hierarchical organization of neural timescales is preserved in humans[11,14]. Given all this work, we argue that there is an overlap in how neural timescales at rest are organized across modalities and species. Nevertheless, the exact correspondence between neuronal spiking, LFP, and BOLD timescales and the extent of their overlap remains unclear. Likewise, the dynamics of neural timescales across modalities and species remain an empirical question.

In conclusion, we demonstrate that neural timescales vary with task engagement that is an aggregate of behavioral state parameters broadly encompassing foraging-related behaviors, action, and spatial navigation encoding. Despite the magnitude of neural timescales expanding with task demands and engagement, we found a stable hierarchical ordering of the areas' neural timescales. We not only provide evidence for the context-dependence of neural timescales, but we also demonstrate that these temporal dynamics were complex and behaviorally relevant. Further investigations and careful experimentation that manipulate the temporal scales over which an animal must integrate information are needed to better understand the link between neural timescales and the temporal scales over which behavior unfolds.

## Methods

### Surgical procedures

Animal procedures were designed and conducted in compliance with the Public Health Service's Guide for the Care and Use of Animals and approved by the Institutional Animal Care and Use Committee of the University of Minnesota. Two male rhesus macaques (*Macaca mulatta*) served as subjects (Age: 7-8). Animals were habituated to laboratory conditions, trained to enter, and exit the open arena, and then trained to operate the water dispensers. We placed a cranial form-fitted Gray Matter (Gray Matter Research) recording chamber and a 128-channel microdrive recording system (SpikeGadgets) over the area of interest. We verified positioning by reconciling preoperative MRI as well as naive skull computed tomography images (CT) with postoperative CTs. Animals received appropriate analgesics and antibiotics after all procedures. The planning of the chamber and subsequent image alignment was performed in a 3D slicer. Brain area segmentation followed the macaque D99 parcellation in NMT space (Saleem et al., 2021)

### Electrophysiological recordings

We recorded with a 128-channel microdrive system (Gray Matter Research), targeting a wide swath of the prefrontal cortex ranging from OFC to PM, and the striatum. Each electrode was independently moveable along the depth dimension. Neural recordings were acquired with a wireless data logger (HH128; SpikeGadgets). The data logger was triggered to start recording with a wireless RF transceiver and periodically received synchronization pulses. Data were recorded at 30 kHz, stored on a memory card for the duration of the experiment, and then offloaded after completion of the session. Each reward station had a local code running the experiment. Task events triggered a TTL pulse, as well as a wireless event code. A dedicated PC running custom code controlled all reward stations, and aggregated event codes. Syncing of all data sources was accomplished via the Main Control Unit (MCU; Spikegadgets), which received dedicated inputs from the pose acquisition system (see below), and reward stations.

Recording sessions were initiated and controlled by Trodes software (Spikegadgets). After neural recordings were offloaded, they were synced with other sources of data via the DataLogger GUI (Spikegadgets).

We recorded for 4-6 days weekly for a period of 4-6 months. For an initial period of 2-4 weeks, we lowered up to ten electrodes in each session until each had punctured the dura and their position was well-within cortex as confirmed from the MRI reconstruction. Subjects still performed experiments, but as the signal was noisy, no recordings were performed during this time. A typical recording day consisted of multiple stages, including electrode adjustment, an experimental session, and extraction of the recorded signal. For the duration of the experiment, on each day, we tracked yields on each electrode and visually assessed the quality of the signal. If an electrode had poor yields for up to 5 days in a row, we would lower it up to 1 mm (or more if it was intended to move to a new area).

All channels were subjected to a custom spike sorting pipeline (modified WaveClust). Channels that exhibited reliable data quality for spike sorting (manual inspection of each channel, including ISI, waveform, and amplitudes >1.7xSNR) were then included in the LFP preprocessing pipeline. To obtain LFPs, we bandpass filtered the raw signal with a second order, two-pass Butterworth filter and Hand taper in the range [0.1300] Hz. Only recordings that showed evidence of neural unit activity (confirmed with separate modified spike sorting analysis) were used for further analysis (see Supplementary Fig. 1 for the number of channels/area). Subdivisions of the brain were collapsed to anatomical areas, listed below as defined in the D99 parcellation of the NMT atlas (Saleem et al., 2021): ACC: 24a', 24a, 24b, 24b', 24c, 24c'; VLPFC: 45a, 45b, 46d, 46v, 46f, 12r; DLPFC: 8bd, 8bs, 9d, 8bm, 9m; FEF: 8ad, 8av; SMA: F3, F6; PM: F1, F2, F5, F7, F4; OFC: 13b, 13m, 13l, 12l, 12m, 12o, 11l, 11m; Striatum.

## Behavioral tracking

We developed a system that can perform detailed three-dimensional behavioral tracking in rhesus macaques with high spatial and temporal precision[30]. The system uses 62 cameras positioned around a specially designed open field environment (2.45 × 2.45 × 2.75 m) with barrels (four barrels located in the corners; height: 78.8 cm; diameter 46.5 cm) in which macaque subjects can move freely in three dimensions and interact with computerized reward stations.

## Pose acquisition and reconstruction

A detailed protocol of the pose acquisition and reconstruction preprocessing can be found here[32].

## Behavioral task

The environment contained four reward stations ("patches") that dispensed water with a programmed delivery schedule. The reward stations were rectangular white boxes with a display monitor placed in the middle, a lever to the left, and a waterspout to the right. The display monitor indicated the availability of the station for foraging (solid blue), reward delivery (solid white background with a solid green cross), or unavailability of the station (i.e., the timeout period; solid white). Each station delivered a fixed amount of water (1.5 ml) per lever press. At any given time, each of the first four lever presses were rewarded and the fifth lever press led to a 3-min timeout period (i.e., depleted station). The subjects could freely decide when and how to interact with the reward stations. No reset or deactivation was applied if the animal left the patch. The timeout could only be triggered after four rewarded and one unrewarded lever press. Each rewarded lever press followed the same programmed sequence. The availability of the reward station was indicated by a solid blue display. A lever press changed the display to white with a green cross in the center, the auditory cue was played, and the solenoid opened to dispense reward. After dispensing, the solenoid closed, the auditory cue ended, and the

green cross disappeared. The display remained white for two additional seconds before it turned blue again. The fifth lever press was instead followed by the screen immediately turning white, with no visual or auditory reward cue and no reward delivery. Other than the interaction with the reward stations, the measured behavior was the subject's unconstrained movement.

## Behavioral variables

**Speed of movement.** For this analysis, we used the 3D center-of-mass (defined as the midpoint between the hip and neck joint) trajectories. We calculated speed as the magnitude of the numerical derivative of the 3D center-of-mass of the subject. We used the tracking data from 166 recording sessions.

**Task-free trials.** Task-free trials were operationalized as windows longer than 5 s when the animal was relatively immobile (i.e., the 3D center-of-mass displacement was less than 40 cm). Task engagement (i.e., continuous interaction with a reward station) was excluded.

**Lever presses.** We recorded behavioral data for 197 sessions (Monkey W: 102; Monkey Y: 95). One session (Monkey Y) was excluded from further analyses since there was no reward station interaction. We divided lever presses into three categories: first, intermediate, and final lever presses. The first lever press was operationalized as the first interaction with a reward station after the animal changed stations. The average number of lever presses per session for this category was 44.7 (Monkey W: 38.7; Monkey Y: 51.1). The final lever press was operationalized as any interaction with a reward station before changing stations. The average number of lever presses per session for this category was 45.1 (Monkey W: 39.1; Monkey Y: 51.4). We took advantage of the fact that, at times subjects prematurely abandoned the reward station after the fourth lever press. Specifically, we compared responses on the fourth lever press when the subject decided to leave versus when they decided to stay for a fifth lever press. The average number of lever presses per session for the stay category was 19.1 (Monkey W: 18.9; Monkey Y: 19.3). The average number of lever presses per session for the leave category was 11.9 (Monkey W: 8.9; Monkey Y: 15.4). The stay lever presses fit within the intermediate category.

## Timescales estimation

**PSD.** PSDs were estimated using the conventional Welch's method, where short-time windowed Fourier transforms are computed from time series and the mean is taken across time. We used 1 s long Hamming windows with 500 ms overlap. To examine whether the Hamming window size has an effect on the areas' relative position in the hierarchy, we also estimated the PSDs using 500 ms (250 ms overlap) and 1500 ms (750 ms overlap) for 20 recording sessions where data was available for all areas (Monkey W: 10; Monkey Y: 10). We did not find any evidence for the Hamming window size having an impact on the areas' relative position in the hierarchy (see Supplementary Fig. 6).

**Spectral parametrization.** Spectral parameterization[41] was applied to extract timescales from PSDs[16]. Briefly, the log-power spectra are decomposed into a summation of narrowband periodic (modeled as Gaussians) and aperiodic (modeled as a Lorentzian function centered at 0 Hz) components. To infer timescales, the periodic components are discarded, and timescales are inferred from the aperiodic component of the PSD. Specifically, $\tau$ can be estimated from the parameter $k$ as $\tau = \frac{1}{2\pi f_k}$, where $f_k \approx k^{1/x}$ is approximated to be the knee frequency, at which a "knee" in the power spectrum occurs (note: equality holds when $\chi = 2$). For a detailed mathematical description of the model and the timescale inference technique, see[16,41], The FOOOF algorithm (version 1.0.0) was used to parameterize neural power spectra.

Settings for the algorithm were set as peak width limits: [2 8]; max number of peaks: 3; minimum peak height: 0; peak threshold: 2; and aperiodic mode: "knee". Power spectra were parameterized across the frequency range 1 to 100 Hz.

**Neural timescales.** Timescales were inferred for each channel individually and then collapsed across channels within an area by taking the median to limit the impact of outliers. We excluded windows for which the PSD parameterization failed or for which the model fit (i.e., R squared) was lower than 0.8. Channels with a failure rate higher than 20% were excluded from further analyses. For an overview of the model fits and the percentage of discarded models per analysis, see Supplementary Fig. 2 and Supplementary Table 1.

**Session-wide.** Two sessions for Monkey Y, and three sessions for Monkey W were excluded due to insufficient time points (i.e., the recording session was less than 95 min). Timescales were inferred by applying spectral parameterization (see above) to the entire session, using a 10 s moving window with a 5 s overlap. To ensure an equal number of windows for each session, only the first 1190 windows were considered for this analysis (i.e., first ~99 min).

**Event-related.** Timescales were inferred 15 s before and after the interaction with a reward station. We inferred timescales by using a 5 s moving window with 2.5 s overlap. We excluded windows for which the PSD parameterization failed or for which the model fit (i.e., $R^2$) was lower than 0.8. To maintain the integrity of the time series associated with each event, the channels with incomplete data (i.e., where at least one window needed to be excluded) were excluded from further analyses. To obtain one event-related time series per session, we subsequently took the median of the resulting neural timescales time series within individual areas. As a result, for each session, we obtained one neural timescales time series per area.

**Task-free.** Timescales were estimated across time over the length of the task-free windows by using a 5 s moving window with 2.5 s overlap over the length of the trial. We excluded windows for which the PSD parameterization failed or for which the model fit (i.e., $R^2$) was lower than 0.8. To obtain one value per trial, we subsequently averaged the neural timescales estimated for any given task-free trial. For each session, the median task-free neural timescales at rest were used as the baseline for further analyses.

**Statistical analysis.** We used the Pearson correlation coefficient to estimate the relationship between the number of lever presses, speed, and neural timescales across individual recording sessions.

Given that neural timescales were not normally distributed, we opted for nonparametric tests to assess statistical significance. We used Wilcoxon signed-rank tests to assess the statistical significance within individual areas—i.e., the change from resting-baseline and the change in neural timescales between neighboring time points. We used the Mann–Whitney U-test to assess the statistical significance between brain areas—i.e., the difference in resting-baseline (or task-free timescales) and the difference in change from resting-baseline around the events of interest. To correct for multiple comparisons, we applied Bonferroni correction. Importantly, for all statistical analyses, we employed the individual sessions as observations rather than the individual events. This was done to mitigate several methodological issues: (1) the different number of events per monkey; (2) the different number of channels per area per monkey and/or event.

**Multiple regression analysis and subsampling.** To perform this analysis, we only included sessions for which the lever press events, behavioral tracking, and neural timescales were available. The total number of data points per area can be calculated as the number of sessions × number of segments (i.e., 10). To quantify the effect of task engagement (i.e., the number of lever presses in any given segment) and the movement of speed on neural timescales, we used a subsampling procedure to estimate the average effect and the confidence interval of these predictors. For each area, we randomly sampled without replacement n (i.e., equivalent to the number of sessions) observations out of the total number of data points. For each subsample, a linear regression model was fitted, with the number of lever presses and speed as regressors, and neural timescales as the response variable. For each area, we repeated the procedure 1000 times. For each predictor, we assessed the difference between areas using an independent sample t-test. For each area, we assessed the difference between predictors using a paired t-test. To correct for multiple comparisons, we applied Bonferroni correction.

**Bayesian regression model.** To assess the relationship between brain areas and session-wide neural timescales, we modeled the predictor (i.e., brain area) as a monotonic effect. This approach is advantageous for ordinal predictors, in this case, the hierarchical organization of brain areas, without falsely treating them as continuous, unordered categorical variables or ordered categorical variables with equidistant levels. In this approach, one estimates one parameter (b) which captures the direction and size of the effect—i.e., the average increase/decrease in the dependent variable associated with the variable. Additionally, one estimates the percentages of the overall increase/decrease that is associated with each of the differences between neighboring variable levels - and hence, these parameters determine the shape of the monotonic effect. For a more detailed explanation, see ref. 42. Brain area was modeled as a monotonic effect and session-wide neural timescales served as the dependent variable.

**Reporting summary**
Further information on research design is available in the Nature Portfolio Reporting Summary linked to this article.

## Data availability
The neural timescales generated in this study have been deposited in the Dryad database under accession code https://doi.org/10.5061/dryad.8sf7m0cx1. The dataset analyzed during the current study is available from the corresponding author on reasonable request. Source data are provided with this paper.

## Code availability
All code and toolboxes used in this study are readily available and cited in the manuscript.

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

## Acknowledgements

We thank the Hayden/Zimmermann lab for valuable discussions and for help with animal care and preparation. This work was supported by NIH grants R01 MH128177 (JZ), P30 DA048742 (JZ, BH, and AZ), R01 MH125377 (BH), NSF 2024581 (JZ and BH) and a UMN AIRP award (JZ, BH, and AZ) from the Digital Technologies Initiative (JZ), from the Minnesota Institute of Robotics (JZ).

## Author contributions

Ana M.G. Manea: Conceptualization, Formal analysis, Investigation, Methodology, Software, Visualization, Writing—original draft, Writing—review and editing. Anna Zilverstand: Writing—review and editing. David J-N Maisson: Data Collection, Data curation. Benjamin Voloh: Data Collection, Data curation. Benjamin Hayden: Conceptualization, Funding acquisition, Methodology, Resources, Writing—review and editing. Jan Zimmermann: Conceptualization, Data curation, Formal analysis, Funding acquisition, Investigation, Methodology, Resources, Software, Supervision, Writing—original draft, Writing—review and editing

## Competing interests
The authors declare no competing interests.
