## [Peer Review File · Nature Communications]

Neural timescales reflect behavioral demands in freely moving rhesus macaquesREVIEWER COMMENTS

Reviewer #1 (Remarks to the Author):

Overview

In this project, Manea et al investigate neural timescales as computed from LFP data collected from freely moving macaques during a foraging task. The goal is to investigate timescales across multiple areas while subjects are engaged in more naturalistic behavior. This is a timely, interesting, and relevant research question to investigate, and a worthy topic to contribute to the literature. By using spectral parameterization of the LFP PSDs, they compute timescales, and examine these across sessions and around events of interests, finding that timescales are dynamic in ways that seem to relate to task engagement, generally, and to specific task events. Overall, I find this an interesting analysis that is a useful contribution to thinking about timescales across the brain in general, and in particular in the quite new approach of doing so by computing them from the LFP data. However, I do find that there are some key conceptual and methodological questions that should be addressed, as well as some lacking information in the manuscript that should be added before it is ready for publication.

Comments

One key conceptual issue that relates to how some of the results are interpreted relates to how this paper refers back to and compares the current results with previous literature. In particular, I am not clear on the way this paper claims that differences in the current results relate to task context, as there is a lack of clarity on this comparison, as well as a lack of addressing potential differences between modalities. For example, in the results, on page 11, it is stated in the "Neural timescales are variable" section, that "the hierarchy of neural timescales is different from previous findings, and hence dependent on the experimental paradigm". It needs to be qualified exactly what this statement is being based on, and how the comparison was made, as I am unclear how this statement is justified. While Figure 1 does report values from other work, these findings are from a different data modality (spiking data, as compared to the field data used here). As such, differences between the results reported here, and from previous work *that estimates a different quantity* could be due to either the modality difference or the task difference (or both). Thus, this difference by itself can not be used to definitively infer a difference that is specific to experimental paradigm, as it seems to be claimed in this work. Indeed, estimated timescales from the different modalities of LFPs vs spikes are known to have different timescales, as is noted in the paper (in some places the current paper does mention differences between modalities, but insufficiently to address this point in my view).

To clearly qualify the statement that the difference in timescales observed in this paper is clearly due to the task context, a clear comparison would have to be made to previous work examining timescales *of LFP data*, however this does not seem to be what was done. In so far as comparisons are made to findings from spiking data, a much clearer discussion (and/or limitations note) is needed to establish the relationship between these measures, the differences observed, and if or how these differences can (or can not) be associated to differences in task context (and/or to differences in recording modality, model species, etc). This issue continues in the discussion wherein other than some brief mentions of modality at the end, much of the discussion treats quite different experiments as if they can be directly compared, which I do not find to be clearly motivated or established. Ultimately, this leads to me being unclear why the ventro-dorsal hierarchy of neural timescales is claimed to be "unique" to the foraging environment (as claimed in the summary and the first paragraph of the discussion), since I neither see a clear reporting of the precise comparisons and results that led to this claim, nor see why the kind of comparison that appears to have been made can be understood to have this kind of specificity, and not be potentially explained by several other possible differences.

In terms of the dataset, there is a lack of reported detail about the data that is analyzed, including the total number of sessions, number of sessions per monkey, average duration of each session, number of electrodes per region, number of lever press events per category (first, final, stay) per analysis, etc. Without having a clearer understanding of the dataset, it is harder to understand

how much data contributes to these analyses and how stable they should be interpreted to be.

There is also no discussion of pre-processing steps and data quality. How many sessions of those recorded “showed evidence of neural unit activity” (p.35) and were thus analyzed, and does this procedure require unit activity across all analyzed areas, or just any unit activity? More detail on the selection criteria that led sessions to be included is needed. Additionally, was any pre-processing applied to, for example, identify bad channels, remove line noise, detect bad data segments, etc? If yes, it should be described, and if not, can we be confident that the data is of consistent quality across monkeys, sessions, and areas?

The methods section that describes the spectral parameterization is lacking in detailed reporting of the model settings and details that is required to understand how the model fitting was done. There are detailed notes associated with the tool in terms of what should be reported: <https://foof-tools.github.io/foof/reference.html>

Additionally, I think a bit more should be said on the data and quality control measures employed related to ensuring that accurate spectral models were fit, such that the inferred timescale can be trusted. Notably, the methods do note that model fits below an R^2 of 0.8 were excluded, however nothing is stated detailing the fit quality of models that were analyzed, nor if this varied across regions, state, or time in a way that might impact the reported measures. Specifically, the average R^2 of kept models and number of dropped models should be reported, for each analysis, as well as checks for whether the R^2 or number of kept models systematically changed across region, time bin, etc. If, for example, the average data quality of task vs baseline data was significantly different, or the number of kept models drastically changed across time bins because of ERP-like activity after events of interest, this could jeopardize the comparisons that are being made and bias the reported results.

I wonder if the authors can also comment on some methodological points related to the event-related analyses (Figures 3 & 4). Based on the methods, the estimates are based on 5 seconds windows, with 50% overlaps, from -15 to +15 seconds. Standardly, this would include 11 windows across such a range, however, here there is a “missing” window: there are 10 windows used total, 5 before 0, and 5 after 0, whereby the window that would be centered +/- 2.5 seconds around 0 (if windows were continuous) is not analyzed or visualized. My worry is if this creates some issues in the analyses and visualizations of this data. For example, in the plots, while it looks like the biggest difference is between the windows before and after zero, this is not a fair comparison to all other windows - since each other pair of adjacent windows actually shares a segment of data (due to the 50% overlap), whereas (due to the “missing” window), the pre/post 0 windows reflect entirely non-overlapping data. Because of this, these two points will almost tautologically be the most different from each other, but not necessarily in an interesting way. For the statistical analyses I was also wary whether this might create some issues with some comparisons (with different adjacent points having different amounts of overlaps), though I did not notice anything in the relevant stats that is clearly an issue. In response to this the authors should at least comment on why they chose this particular approach and why they do not think it is an issue for the figures or statistical analyses. At least, the methods should note these decisions, as the current statement text that implies continuous windows from -15 to +15 seconds is not quite accurate.

I also find several aspects of the figures to be somewhat unclear and insufficiently described in the figure legends:

- For several figures, including 1E, 2C, 2D, 3B, etc: how were the areas ordered on the x-axis in this graph? They look very structured - at first I thought the data points were ordered by an independent variable (such as anatomical location) such that the x-ordering was meaningful. However, when comparing across different panels, it appears that each plot is independently re-ordered based on the y-values, such that the x-ordering has no particular information? I think the ordering that is used should be defined in the figure legend. I also think it would be useful to maintain the same order across plots (choosing some standard order), to make it easier to compare across panels.

- o If I understand correctly, in Figure 2E, the ordering of the areas is by anatomy, ordered from ventral to dorsal areas? If so, I would suggest maintaining this order for all plots across areas - this seems like it would greatly assist in comparing across measures, and evaluating when

measures do follow the anatomical hierarchy (and when they don't).

o As is, a panel like 3B is a bit confusing, since the legend states that the timescales are hierarchically organized, but the graph seems to be ordered by value. I think the regions above show the anatomical order of the regions - but this is not described.

- Figure 2A: based on the methods (page 12), this plot reflects estimates based on a fixed 10 second window that is stepped along, with the x-axis indicating start time of each window with respect to the session time, right? Based on the figure description, I feel like this is not particular clear from the figure legend and could be clarified. For example, before checking the methods, I thought the x-axis might reflect cumulative time (the first data point includes the first minute of data, whereas the last point includes 100 minutes of included data in the window, etc).

- Figure 2B: frequency units need to be specific - is it Hz? Frequency per minute? It's unclear. The legend says "per time bin", but this also is unclear, as it does not seem like it can be the 10 seconds windows that are being referred to, and yet no other time bin definition is stated.

- Figure 2C-D: given that a key goal of presenting these two set of correlations is (I think) to show that the level press correlation >> speed correlation, this might be clearer if the y-axis were put on the same scale?

- Figure 2E: I think it would be useful to use a color contrast for task engagement vs speed that are not colors used to label areas - using different transparency or textures while keeping the same area color scheme might be better (so that a "red" region would be visualized as red dots with high & low contrast or with & without stripes, to match other panels). This goes for Figure 5 as well.

- Figure 4: it is not clearly stated what the values in the triangular plots are. I initially assumed they might be absolute or normalized differences, but based on the values / shading it looks like they are p-values? The legend should be clarified. I also think reporting a difference measures or effect size might be more informative.

- Figure 4: In the line plots of B & C, I am unclear what the circles (in dashed lines) around groups of regions at the -1 and +1 time points reflect.

- Figure 4: In the line plots of B & C, the x-axis labels are not accurate - the label says "time (s)", but the colored numbers are actually window indices (right?) and do not reflect seconds. Alternately stated, the plot as is looks like it ranges from -5 to +5 seconds, whereas the actual time range is -15 to +15. Either the axis label needs to be updated, or the values in seconds (as seen in the right of panel A) need to be added.

- Figure 5: titles above the plots in B-E would be useful to be able to interpret these plots without needing to constantly check which is which from the legend

Some additional minor comments:

p6 - "even sheer neuronal variability is ubiquitous and functional" - I find this phrasing unclear as to what it's trying convey.

p11 - first sentence of "Neural timescales are variable" repeats word "is"

Reviewer #2 (Remarks to the Author):

Neural timescales reflect behavioral demands in freely moving rhesus macaques

Manea et al. present findings from two macaques foraging in an open environment while neural recordings are obtained from 8 distinct brain areas using a wireless neural logger system. In this specific study, the authors ask whether "neural timescales" vary compared to more restrained laboratory environments; as well as a series of other questions such as whether they vary across brain areas, time, and task engagement. If findings in this unrestrained environment are significantly different from restrained laboratory tasks, this could be an important piece of information justifying, perhaps, more unrestrained paradigms in the future. I also note the strength of their video-based tracking, which has been the object of previous publications by the same group.

I have a major concern about the method used in this paper and how it supports its conclusion. Namely, all other laboratory tasks cited for comparison in the manuscript and included in Figure 1A

used “spike-counts” to calculate the decay of the autocorrelation function, aka. the neural timescale. However, in this paper by Manea et al. the authors used local field potentials rather than spike-counts and state: “We demonstrated that the hierarchy of neural timescales is (1) is different from previous findings, and hence dependent on the experimental paradigm used”. This major conclusion could either be supported by the thesis of the authors (e.g. restrained vs. unrestrained paradigms), or conversely by the rather trivial fact that the authors used a different neural metrics, namely LFPs, to draw their conclusions. Unfortunately, I have not seen in the paper convincing evidence that these two (i.e. LFPs and spike-based metrics) are comparable. I am also unconvinced by the justification they provided: “we focused on the LFPs because of their spatial coverage”. I also wonder why the authors didn’t not use the spikes they have recorded from these areas (which they mention having in their possession in the method) to run their analyses. It would have been a much stronger study with direct relevance to the existing literature.

I also felt the abstract and introduction were long and convoluted. Moreover, it is assumed the reader is familiar with the concepts of “hierarchies of neural timescales” from the very first sentence of the abstract. I strongly encourage taking a sentence or two to define this first. I would recommend shortening the intro and getting to the point of why this metric matters. I liked the 4th paragraph which motivates the research question, but felt like the first 3 paragraphs could be condensed into one or two, making this an easier read.

I’m sorry to say that I did not feel an excitement for those results commensurate with the prestige of the journal they are submitted to. The use of a highly-derived, post-processed neural metric with unknown functional relevance in the context of a purely correlational paradigm did not strike my curiosity. This may simply reflect my ignorance towards this subject however, and other reviewers may beg to differ.

Reviewer #3 (Remarks to the Author):

This manuscript studies how timescales of LFP activity relate to behavioral variables in the context of foraging behavior in freely behaving monkeys. The results suggest that neural timescales during foraging are hierarchically organized in an order that overlaps with the hierarchical organization of timescales measured during structured-constrained experiments but also has some differences. Moreover, the study demonstrates that while preserving the hierarchical order, neural timescales increase with task engagement and correlate with specific foraging events.

I find studying neural timescales during a natural foraging task instead of constrained experiments fascinating, especially the fact that the hierarchical order even during the task-free periods differs from previous studies using constrained-structured experiments. The findings are interesting and provide new insights into the role of neural timescales in brain function and computation. However, there are some logical and technical concerns that I have outlined below.

1. The framing of the introduction and comparison with the previous work:

1. a. In the introduction, the authors claim “There is currently conflicting evidence about the behavioral dependence of neural timescales, their hierarchical organization and general function” and as evidence for this claim, they refer to studies that show timescales can change with task demands and compare them with studies that found a hierarchical organization of timescales across the brain at rest. However, I do not see any controversy between these two sets of findings. Having a replicable hierarchical organization at rest does not exclude the possibility that timescales can change during task performance. In fact, Gao et al., 2020 provided evidence for the existence of both phenomena in the same data: the hierarchical organization of timescales within association regions exists during the baseline period and all regions exhibit a ~20% increase in their timescales after engagement in the working memory task.

1. b. In the introduction, the authors claim “ We propose that the failure to find multiple hierarchies of neural timescales in previous experiments may be a by-product of the rigid structure of traditional experiments that enforce stationary temporal scales.”, but it is not clear what they mean by “multiple hierarchies”. Previous studies such as Spitmaan et al., 2020 have found multiple

hierarchies of timescales (i.e. intrinsic and task-related timescales) that all follow a similar order of brain areas. Authors might mean that all previous studies suggested a similar order of brain areas for the hierarchy, in that case, it is better to clarify.

1. c. For readers to better contextualize the results, I think it is better to unpack what the authors mean by "In summary, we found a ventro-dorsal hierarchy that partially overlapped but also deviated from previous findings in important ways" in the results section. Detailed explanations for this difference are now provided in the discussion, but having some sort of visualization that compares the order of brain areas in the previous studies and the current study is helpful in understanding the results.

2. Measuring timescales and regression analysis:

2. a. For measuring timescales authors consider two types of windows: (i) Hamming window of size 1 sec for computing PSD, (ii) Sliding window of size 10/5 sec (depending on the analysis) to compute time-dependent timescales. It is important to show that the results presented in the paper are not dependent on the chosen window sizes, or at least show within what range of window sizes the results are replicable. This is particularly important since the size of Hamming window defines the largest estimated timescale and the size of the timescale window (if too larger) might hinder some changes in the neural timescales. It is also not clear why the authors chose a different window size for the analysis in Fig 2A, compared to the rest of the paper.

2. b. Regarding "To assess the significance of this linear increase for individual areas, we conducted a linear regression model for each event with time as the predictor and neural timescales as the dependent variable." (page 18), details of the analysis are unclear and do not quite match the visualizations in Fig 3 C, D. The regression analysis should potentially be done using data around all individual events (across sessions/monkeys) and not the summary of data (depending on the variability across sessions, separate intercepts might need to be fitted). However, in Fig 3 C, D, it seems some summary of data (unclear whether mean/median) without any error bars are presented. A proper visualization for the regression analysis, would be to plot all the measured timescales (around each event across sessions/monkeys) relative to the time to/from the event together with the fitted regression line. Moreover, the parameters of regression analysis (e.g., the slope), number of data samples and R-squared need to be reported (e.g., in a Table), since significance alone might not indicate strong correlations between timescales and their relative temporal distance to the events.

3. Hierarchy of change in the magnitude of neural timescales relative to the events (Fig 4, 5):

3. a. The amount of change in the magnitude of timescale across brain areas is computed as the difference to the baseline (task-free) but this comparison is more meaningful if the difference is also normalized to the baseline of each area, i.e. $(\tau(\text{task}) - \tau(\text{baseline})) / \tau(\text{baseline})$. This relative normalization ensures that the structure presented in Fig 4, 5 is not a mere consequence of brain areas with larger timescales having a larger absolute change in their timescales and hence providing no additional insight to the already observed hierarchy in the magnitude of timescales (for instance, it might be the case that all areas are increasing their timescales by the same percentage suggesting that a global mechanism is modulating the timescales across all areas).

3. b. It is difficult to draw clear conclusions from all the changes presented in Fig 4, 5, providing a summary figure with the main conclusions (considering only statistically significant results) would be helpful.

Minor comments and clarifications:

- Please add line numbers to the manuscript.

- It seems the authors are using the phrases "segments/bins" interchangeably (e.g., Fig 2C), it would be nice to make it consistent across the text using one of the two phrases. Please also mention the size of the bin in the caption of Fig 2C, D.

- In the caption of Fig 3B-D and Fig 4B, C, please mention what dots (and error bars if plotted) indicate (mean/median/etc), and how are they computed (across sessions/monkeys/etc). Please also explain what the dashed circles represent in Fig 4.

- For figures that you computed p-values, please add the number of samples to the caption and for Bonferroni corrections please mention for how many multiple comparisons.

References

Gao, Richard, et al. "Neuronal timescales are functionally dynamic and shaped by cortical microarchitecture." *Elife* 9 (2020): e61277.

Spitmaan, Mehran, et al. "Multiple timescales of neural dynamics and integration of task-relevant signals across cortex." *Proceedings of the National Academy of Sciences* 117.36 (2020): 22522-22531.

Response to reviewers:

Neural timescales reflect behavioral demands in freely moving rhesus macaques

Ana M.G. Manea^{1,2}, Anna Zilverstand³, David J.-N. Maisson, Benjamin Voloh, Benjamin Hayden⁴, Jan Zimmermann^{1,2}

Affiliations

1 Department of Neuroscience, University of Minnesota, Minneapolis MN

2 Center for Magnetic Resonance Research, University of Minnesota, Minneapolis MN

3 Department of Psychiatry and Behavioral Sciences, University of Minnesota, Minneapolis MN

4 Department of Neurosurgery, Baylor College of Medicine, Houston, TX

Reviewer 1:

R1.1. Comment: One key conceptual issue that relates to how some of the results are interpreted relates to how this paper refers back to and compares the current results with previous literature. In particular, I am not clear on the way this paper claims that differences in the current results relate to task context, as there is a lack of clarity on this comparison, as well as a lack of addressing potential differences between modalities. For example, in the results, on page 11, it is stated in the “Neural timescales are variable” section, that “the hierarchy of neural timescales is different from previous findings, and hence dependent on the experimental paradigm”. It needs to be qualified exactly what this statement is being based on, and how the comparison was made, as I am unclear how this statement is justified. While Figure 1 does report values from other work, these findings are from a different data modality (spiking data, as compared to the field data used here). As such, differences between the results reported here, and from previous work *that estimates a different quantity* could be due to either the modality difference or the task difference (or both). Thus, this difference by itself can not be used to definitively infer a difference that is specific to the experimental paradigm, as it seems to be claimed in this work. Indeed, estimated timescales from the different modalities of LFPs vs spikes are known to have different timescales, as is noted in the paper (in some places the current paper does mention differences between modalities, but insufficiently to address this point in my view). To clearly qualify the statement that the difference in timescales observed in this paper is clearly due to the task context, a clear comparison would have to be made to previous work examining timescales *of LFP data*, however this does not seem to be what was done. In so far as comparisons are made to findings from spiking data, a much clearer discussion (and/or limitations note) is needed to establish the relationship between these measures, the differences observed, and if or how these differences can (or cannot) be associated to differences in task context (and/or to differences in recording modality, model species, etc). This issue continues in the discussion wherein other than some brief mentions of modality at the end, much of the discussion treats quite different experiments as if they can be directly compared, which I do not find to be clearly motivated or established.

R.1.1. Response: We thank the reviewer for this insightful comment and apologize for the lack of clarity on our behalf. This point was also raised by the second reviewer. In this manuscript, we are trying to make two separate but equally important arguments. The first argument pertains to the hierarchical organization of neural timescales, and its task-dependence. When it comes to the hierarchy being task-dependent, we agree with the reviewer that the results reported here could either be explained by the task difference, the modality being used or both. However, Gao et al. (2020) show that neural timescales estimated from ECoG display the same hierarchical organization as those estimated by Murray et al. (2014) with spiking data ($\rho = 0.95$, $p < 0.001$). Importantly, they replicate the spiking results with an independent ECoG dataset. Indeed, ECoG timescales are ~10x faster than the spiking neural timescales but the areas' relative position in the hierarchy is consistent. We believe that this is strong evidence that although the absolute value of the estimated neural timescales is modality specific, the hierarchical organization does not change. Moreover, we have previously shown that neural timescales estimated from neuronal spiking and fMRI, a modality with high correspondence to LFP, display a very similar hierarchical organization in macaques (Manea et al., 2022). Similar to the spiking-LFP correspondence, hemodynamic neural timescales are slower (seconds) than spiking timescales (ms), however the areas' relative position in the hierarchy is consistent. All these comparisons have been made based on neural timescales at rest. Moreover, this hierarchical organization of neural timescales at rest seems to be preserved across species as well, although there is no study yet that systematically makes this comparison. In particular, there is evidence that the unimodal areas to association areas axis of hierarchical organization of neural timescales is preserved in humans (Ito et al., 2020; Raut et al., 2020). In conclusion, the literature supports the claim that the

hierarchical organization of neural timescales at rest is preserved across species and modalities. Therefore, we argue that the observed hierarchy in this study could arise from having the animals in a completely different environment, with very different input and output demands. That being said, we do agree with the reviewer that direct comparisons across modalities and species are needed.

The second argument pertains to the modulation of the neural timescales' magnitude as a function of behavioral demands. This modulation in the magnitude of neural timescales has been previously shown with both human LFP data (Gao et al., 2020) and macaque spiking (Zeraati et al., 2023). Therefore, we argue that this modulation is not modality-specific either.

As the reviewer states, we did not provide a thorough discussion addressing this and have done so now. In summary, we justify the use of LFP data in more detail and argue why we and how the empirical literature suggests that timescale hierarchies should be consistent between measures.

Revised Discussion (page 19-20):

Deviations from the commonly reported hierarchical organization of neural timescales estimated from monkey neuronal spiking have been previously described in a study using human LFP data (Gao et al., 2020). In particular, the OFC displayed slower timescales than other PFC structures, which is the opposite of what had been previously reported (see Fig. 6A for comparisons). It is important to note that our results are not incompatible with the previous literature on hierarchies of neural timescales at rest but are rather complementary by investigating neural timescales in a new context, that of unconstrained behavior. While these differences in the relative position of an area within the hierarchy could be due to using different modalities, previous work demonstrating a strong correspondence between signal modalities (Gao et al., 2020) suggests they are the result of contextual adaptation of the observed hierarchy of neural timescales.

Revised Discussion (page 22):

Here we use LFP rather than single-unit activity to infer timescales for two reasons. First, LFP activity, in our dataset, offered much broader spatial and temporal coverage and provided us with the ability to record a large number of areas simultaneously. That is, we were able to leverage the nature of this signal to infer neural timescales in all areas across time throughout the recording session. Second, the neuronal firing rates in this dataset are sparse and hence, it would have been difficult if not impossible to calculate the autocorrelation function across time for many cells. It is possible that the results reported here are unique to LFP activity and do not fully translate to single-unit activity. Nevertheless, it is important to note that the LFP signal itself is unique in that it is a continuous signal that can be leveraged across long temporal windows allowing us to probe the dynamics of neural timescales. An equivalent analysis could not be easily accomplished with single-unit activity. Our main goal was to use whatever brain signal allows us to investigate the temporal dynamics of neural timescales in this unconstrained foraging context from a population perspective. Notably, even timescales estimated from neuronal spiking have been generally described and reported as a population-level statistic due to their high heterogeneity exhibited at the single-neuron level (Murray et al., 2014). That being said, although LFP and single-unit activity are fundamentally different signals, Gao et al., 2020 showed that macaque ECoG timescales track previously published spiking timescales (Murray et al., 2014). Moreover, we have previously shown that neural timescales estimated from neuronal spiking and fMRI, a modality closely related to LFP (Logothetis et al., 2001), display a very similar hierarchical organization (Manea et al., 2022). There is also evidence that the unimodal areas to association areas axis of hierarchical organization of neural timescales is preserved in humans (Ito et al., 2020; Raut et al., 2020). Given all this work, we argue that there is an overlap in how neural timescales at rest are organized across modalities and species. Nevertheless, the exact correspondence between neuronal spiking, LFP and

BOLD timescales and the extent of their overlap remains unclear. Likewise, the dynamics of neural timescales across modalities and species remain an empirical question.

R1.2. Comment: Ultimately, this leads to me being unclear why the ventro-dorsal hierarchy of neural timescales is claimed to be “unique” to the foraging environment (as claimed in the summary and the first paragraph of the discussion), since I neither see a clear reporting of the precise comparisons and results that led to this claim, nor see why the kind of comparison that appears to have been made can be understood to have this kind of specificity, and not be potentially explained by several other possible differences.

R1.2. Response: We thank the reviewer for raising this important point. The reviewer is correct that the use of the word “unique” in the introduction and discussion could be misleading. We have rewritten the abstract to make sure this is not misunderstood.

We have also made sure that the discussion could not be misunderstood in the same way and replaced the main statement to:

Revised Discussion (page 19):

Here we found a ventro-dorsal hierarchy of neural timescales that is influenced by this foraging environment. Importantly, we showed that this hierarchy is preserved even in the context of flexible task demands.

R1.3. Comment: In terms of the dataset, there is a lack of reported detail about the data that is analyzed, including the total number of sessions, number of sessions per monkey, average duration of each session, number of electrodes per region, number of lever press events per category (first, final, stay) per analysis, etc. Without having a clearer understanding of the dataset, it is harder to understand how much data contributes to these analyses and how stable they should be interpreted to be.

R.1.3. Response: We thank the reviewer for this suggestion. We have thoroughly revised the manuscript to include the following detail as requested. We added **Supplementary Fig. 1** with information about the number of channels per area. We added **Supplementary Fig. 2** which includes information about model fits across time, areas and analyses (A-F). Additionally, this figure includes the number of sessions used across analyses (G-P). It is important to note that in all analyses, rather than using the individual channels or events as observations in our statistical analyses, we averaged the neural timescales to obtain one value per session. As a result, the number of sessions per area was used as our sample size for all analyses (see **Supplementary Fig. 2G-P**).

Revised Methods (page 24-25):

We recorded behavioral data for 198 sessions (Monkey W: 102; Monkey Y: 96). One session (Monkey Y) was excluded from further analyses since there was no reward station interaction. The average daily recording session was 97.8 minutes (SD ± 5.2 minutes). We divided lever presses into three categories: first, intermediate and final lever presses. The first lever press was operationalized as the first interaction with a reward station after the animal changed stations. The average number of lever presses per session for this category was 44.7 (Monkey W: 38.7; Monkey Y: 51.1). The final lever press was operationalized as any interaction with a reward station before changing stations. The average number of lever presses per session for this category was 45.1 (Monkey W: 39.1; Monkey Y: 51.4). We took advantage of the fact

that at times subjects prematurely abandoned the reward station after the fourth lever press. Specifically, we compared responses on the fourth lever press when the subject decided to leave versus when they decided to stay for a fifth lever press. The average number of lever presses per session for the stay category was 19.1 (Monkey W: 18.9; Monkey Y: 19.3). The average number of lever presses per session for the leave category was 11.9 (Monkey W: 8.9; Monkey Y: 15.4).

R1.4. Comment: There is also no discussion of pre-processing steps and data quality. How many sessions of those recorded “showed evidence of neural unit activity” (p.35) and were thus analyzed, and does this procedure require unit activity across all analyzed areas, or just any unit activity? More detail on the selection criteria that led sessions to be included is needed. Additionally, was any pre-processing applied to, for example, identify bad channels, remove line noise, detect bad data segments, etc? If yes, it should be described, and if not, can we be confident that the data is of consistent quality across monkeys, sessions, and areas?

R1.4. Response: We thank the reviewer for this suggestion and very much agree that this information is important. We have now added details on the parameters of datasets and electrodes included as suggested. We have now described our approach in more detail in **Supplementary Fig. 2** and **Supplementary Table 1** as well as the methods section. Because of the size of the dataset we could not visually inspect each channel's LFP over the entirety of the sessions. Moreover, to maintain the integrity of the time series in order to be able to examine the temporal dynamics of neural timescales, we chose not to remove any data segments before parameterizing the neural power spectra. Instead, we opted for a conservative approach. All channels were subjected to a custom spike sorting pipeline (leveraging WaveClust). Channels that exhibited reliable data quality for spike sorting (manual inspection of each channel, including ISI, waveform, amplitudes $> 1.7 \times \text{SNR}$) were then included in the LFP preprocessing pipeline. This is a fairly conservative threshold as linoise or bad channels would not make it past that threshold. Because the animals have a large titanium chamber that acts as a shield and are floating (DC battery logger), data quality was surprisingly clean. Comparable recordings in the animals' chairs suffered from significantly more noise. Lastly, the model fitting approach we take acts as another quality control as segments that could contain noise contamination would be rejected. Lastly, channels which displayed a high rate of model fitting failure or bad fits (i.e., $> 20\%$) were excluded from the analyses. Given that the electrodes were lowered across the recording period, some recording sessions do not have data for all areas.

We added **Supplementary Fig. 2** which includes information about model fits across time, areas, and analyses (A-F). We added **Supplementary Table 1** which includes information about the percentage of discarded models across analyses.

Revised Methods (page 23):

All channels were subjected to a custom spike sorting pipeline (modified WaveClust). Channels that exhibited reliable data quality for spike sorting (manual inspection of each channel, including ISI, waveform, amplitudes $> 1.7 \times \text{SNR}$) were then included in the LFP preprocessing pipeline.

R1.5. Comment: The methods section that describes the spectral parameterization is lacking in detailed reporting of the model settings and details that is required to understand how the model fitting was done. There are detailed notes associated with the tool in terms of what should be reported: <https://foof-tools.github.io/foof/reference.html>

R1.5. Response: We have amended the method section as suggested by the reviewer. We thank the reviewer for this suggestion.

Revised Methods (page 25):

The FOOOF algorithm (version 1.0.0) was used to parameterize neural power spectra. Settings for the algorithm were set as: peak width limits: [2 8]; max number of peaks : 3; minimum peak height : 0; peak threshold: 2; and aperiodic mode : “knee”. Power spectra were parameterized across the frequency range 1 to 100 Hz.

R1.6. Comment: Additionally, I think a bit more should be said on the data and quality control measures employed related to ensuring that accurate spectral models were fit, such that the inferred timescale can be trusted. Notably, the methods do note that model fits below an R^2 of 0.8 were excluded, however nothing is stated detailing the fit quality of models that were analyzed, nor if this varied across regions, state, or time in a way that might impact the reported measures. Specifically, the average R^2 of kept models and number of dropped models should be reported, for each analysis, as well as checks for whether the R^2 or number of kept models systematically changed across region, time bin, etc. If, for example, the average data quality of task vs baseline data was significantly different, or the number of kept models drastically changed across time bins because of ERP-like activity after events of interest, this could jeopardize the comparisons that are being made and bias the reported results.

R1.6. Response: We thank the reviewer for raising this important point. As suggested, we have now added this information. We added **Supplementary Fig. 2**, which includes information about model fits across time, areas, and analyses (A-F). Moreover, we added **Supplementary Table 1**, which includes information about the percentage of discarded models across analyses. Although there is variation in the average model fit and discarded models across analyses, the R^2 of the neural timescales used in our analyses ranged from 0.95 and 0.99. The number of discarded models tends to be higher and R^2 tends to be lower in the event-related estimation. To mitigate this, for any given event (i.e., lever press) we only used the channels with good fits across all time points. This was done independently for before and after the event to avoid excluding too many channels. Another reason for choosing this approach was to have intact before/after event-related “time series” of neural timescales. As mentioned earlier, after including only channels with good fit across all time points, we averaged across channels within anatomical areas and we obtained one time series per event per area for each recording session.

Revised Methods (page 26):

To maintain the integrity of the time series associated with each event, the channels with incomplete data (i.e., where at least one window needed to be excluded) were excluded from further analyses. To obtain one event-related time series per session, we subsequently took the median of the resulting neural timescales time series within individual areas. As a result, for each session we obtained one neural timescales time series per area.

R1.7. Comment: I wonder if the authors can also comment on some methodological points related to the event-related analyses (Figures 3 & 4). Based on the methods, the estimates are based on 5 seconds windows, with 50% overlaps, from -15 to +15 seconds. Standardly, this would include 11 windows across such a range, however, here there is a “missing” window: there are 10 windows used total, 5 before 0, and 5 after 0, whereby the window that would be centered +/- 2.5 seconds around 0 (if windows were continuous) is not analyzed or visualized. My worry is if this creates some issues in the analyses and visualizations of this data. For example, in the plots, while it looks like the biggest difference is between the windows before and after

zero, this is not a fair comparison to all other windows - since each other pair of adjacent windows actually shares a segment of data (due to the 50% overlap), whereas (due to the “missing” window), the pre/post 0 windows reflect entirely non-overlapping data. Because of this, these two points will almost tautologically be the most different from each other, but not necessarily in an interesting way. For the statistical analyses I was also wary whether this might create some issues with some comparisons (with different adjacent points having different amounts of overlaps), though I did not notice anything in the relevant stats that is clearly an issue. In response to this the authors should at least comment on why they chose this particular approach and why they do not think it is an issue for the figures or statistical analyses. At least, the methods should note these decisions, as the current statement text that implies continuous windows from -15 to +15 seconds is not quite accurate.

R.1.7. Response: We thank the reviewer for this insightful comment and agree that our primary motivation for performing the analysis the way we did was not described well in the manuscript. We were particularly interested in changes before and after the event, and hence we time-locked the estimation of the neural timescales to the event to examine how the event itself affects these dynamics. The window centered on the event would include both the dynamics of both before and after this decision has been made and would be harder to interpret. The question of interest with respect to this midpoint is whether the neural timescales display the same trend (increase/decrease depending on the event). We have now clearly stated this in the results section. We have also estimated and plotted the neural timescales time series for the first and final lever presses (see reviewer figure below). Overall, the value for time point 0 is an intermediate between the before and after values.

Revised Results (page 10):

Because we were particularly interested in changes before and after the lever presses, we time-locked the estimation of the neural timescales to the event to examine how the event itself affects these dynamics (Fig. 3A). We excluded the window centered on the event since it would include the dynamics of both before and after this decision has been made.

Figure for reviewers. Neural timescales for events with different behavioral contexts

(A) Neural timescales (τ) estimation for task-free segments and lever presses. **Left:** Neural timescales surrounding the first lever press. Vertical dotted line: time of lever press. Circles: median across sessions \pm s.e.m. **Right:** Neural timescales surrounding the final lever press. Vertical dotted line: time of lever press. Circles: median across sessions \pm s.e.m.

R1.8. Comment: I also find several aspects of the figures to be somewhat unclear and insufficiently described in the figure legends:

- For several figures, including 1E, 2C, 2D, 3B, etc: how were the areas ordered on the x-axis in this graph? They look very structured - at first I thought the data points were ordered by an independent variable (such as anatomical location) such that the x-ordering was meaningful. However, when comparing across different panels, it appears that each plot is independently re-ordered based on the y-values, such that the x-ordering has no particular information? I think the ordering that is used should be defined in the figure legend. I also think it would be useful to maintain the same order across plots (choosing some standard order), to make it easier to compare across panels.

- o If I understand correctly, in Figure 2E, the ordering of the areas is by anatomy, ordered from ventral to dorsal areas? If so, I would suggest maintaining this order for all plots across areas – this seems like it would greatly assist in comparing across measures, and evaluating when measures do follow the anatomical hierarchy (and when they don't).

- o As is, a panel like 3B is a bit confusing, since the legend states that the timescales are hierarchically organized, but the graph seems to be ordered by value. I think the regions above show the anatomical order of the regions - but this is not described.

- Figure 2A: based on the methods (page 12), this plot reflects estimates based on a fixed 10 second window that is stepped along, with the x-axis indicating start time of each window with respect to the session time, right? Based on the figure description, I feel like this is not particular clear from the figure legend and could be clarified. For example, before checking the methods, I thought the x-axis might reflect cumulative time (the first

data point includes the first minute of data, whereas the last point includes 100 minutes of included data in the window, etc).

- Figure 2B: frequency units need to be specific - is it Hz? Frequency per minute? It's unclear. The legend says "per time bin", but this also is unclear, as it does not seem like it can be the 10 seconds windows that are being referred to, and yet no other time bin definition is stated.

- Figure 2C-D: given that a key goal of presenting these two set of correlations is (I think) to show that the level press correlation >> speed correlation, this might be clearer if the y-axis were put on the same scale?

- Figure 2E: I think it would be useful to use a color contrast for task engagement vs speed that are not colors used to label areas - using different transparency or textures while keeping the same area color scheme might be better (so that a "red" region would be visualized as red dots with high & low contrast or with & without stripes, to match other panels). This goes for Figure 5 as well.

- Figure 4: it is not clearly stated what the values in the triangular plots are. I initially assumed they might be absolute or normalized differences, but based on the values / shading it looks like they are p-values? The legend should be clarified. I also think reporting a difference measures or effect size might be more informative.

- Figure 4: In the line plots of B & C, I am unclear what the circles (in dashed lines) around groups of regions at the -1 and +1 time points reflect.

- Figure 4: In the line plots of B & C, the x-axis labels are not accurate - the label says "time (s)", but the colored numbers are actually window indices (right?) and do not reflect seconds. Alternately stated, the plot as is looks like it ranges from -5 to +5 seconds, whereas the actual time range is -15 to +15. Either the axis label needs to be updated, or the values in seconds (as seen in the right of panel A) need to be added.

- Figure 5: titles above the plots in B-E would be useful to be able to interpret these plots without needing to constantly check which is which from the legend

Some additional minor comments:

p6 - "even sheer neuronal variability is ubiquitous and functional" - I find this phrasing unclear as to what it's trying convey.

p11 - first sentence of "Neural timescales are variable" repeats word "is"

R1.8. Response: We thank the reviewer for these constructive comments. We have changed the figures and text to include the insightful feedback. As an extension of Fig. 4, we also added Fig. 6B which includes the average effect sizes for each time point.

Reviewer #2:

R2.1. Comment: I have a major concern about the method used in this paper and how it supports its conclusion. Namely, all other laboratory tasks cited for comparison in the manuscript and included in Figure 1A used "spike-counts" to calculate the decay of the autocorrelation function, aka. the neural timescale. However, in this paper by Manea et al. the authors used local field potentials rather than spike-counts and state: "We demonstrated that the hierarchy of neural timescales is (1) is different from previous findings, and hence dependent on the experimental paradigm used". This major conclusion could either be supported by the thesis of the authors (e.g. restrained vs. unrestrained paradigms), or conversely by the rather trivial fact that the authors used a different neural metrics, namely LFPs, to draw their conclusions. Unfortunately, I have not seen in the paper convincing evidence that these two (i.e. LFPs and spike-based metrics) are comparable. I am also unconvinced by the justification they provided: "we focused on the LFPs because of their spatial coverage". I also wonder why the authors didn't not use the spikes they have recorded from these areas (which they mention having in their possession in the method) to run their analyses. It would have been a much stronger study with direct relevance to the existing literature.

R2.1. Response: We thank the reviewer for this constructive comment. As far as the spikes are concerned, we agree with the reviewer that it would be wonderful for us to use those data for these analyses, but we ran into a logistical and statistical issue in doing so. Spike counts in free behavior in the prefrontal cortex can be extremely low with clear biases towards certain mobility and action related variables. The resulting analysis would thus be extremely biased to temporal intervals of the experiment and hard if not impossible to interpret. If we restricted ourselves to only the higher firing rate cells, we would introduce yet another bias. Using LFPs was therefore the only way for us to assess timescales over longer periods. We hope that in the future, using higher channel count probes, we can have access to stable spike recordings over long sessions to allow the analysis the reviewer is asking for.

We further acknowledge and agree that this hierarchical organization of neural timescales could be the result of using LFP rather than spiking data. However, Gao et al., 2020 showed that macaque ECoG track previously published spiking timescales (Murray et al., 2014). Moreover, we have previously shown that neural timescales estimated from neuronal spiking and fMRI display a very similar hierarchical organization (Manea et al., 2022). Moreover, even timescales estimated from neuronal spiking have been generally described and reported as a population-level statistic due to the extremely high heterogeneity observed at the single-neuron level (see Murray et al., 2014 for example). There is also evidence that the unimodal areas to association areas axis of hierarchical organization of neural timescales is preserved in humans (Ito et al., 2020; Raut et al., 2020). Given all this work, we argue that there is an overlap in how neural timescales are organized across modalities. Nevertheless, the exact correspondence between neuronal spiking, LFP and BOLD timescales and the extent of their overlap remains unclear. Likewise, the dynamics of neural timescales across modalities and species remain an empirical question.

Revised Discussion (page 19-20):

Deviations from the commonly reported hierarchical organization of neural timescales estimated from monkey neuronal spiking have been previously described in a study using human LFP data (Gao et al., 2020). In particular, the OFC displayed slower timescales than other PFC structures, which is the opposite of what had been previously reported (see Fig. 6A for comparisons). It is important to note that our results are not incompatible with the previous literature on hierarchies of neural timescales at rest, but are rather complementary by investigating neural timescales in a new context, that of unconstrained behavior. While these differences in the relative position of an area within the hierarchy could be due to using different modalities, previous work demonstrating a strong correspondence between signal modalities (Gao et al., 2020) suggests they are the result of contextual adaptation of the observed hierarchy of neural timescales.

Revised Discussion (page 22):

Here we use LFP rather than single-unit activity to infer timescales for two reasons. First, LFP activity, in our dataset, offered much broader spatial and temporal coverage and provided us with the ability to record a large number of areas simultaneously. That is, we were able to leverage the nature of this signal to infer neural timescales in all areas across time throughout the recording session. Second, the neuronal firing rates in this dataset are sparse and hence, it would have been difficult if not impossible to calculate the autocorrelation function across time for many cells. It is possible that the results reported here are unique to LFP activity and do not fully translate to single-unit activity. Nevertheless, it is important to note that the LFP signal itself is unique in that it is a continuous signal that can be leveraged across long temporal windows allowing us to probe the dynamics of neural timescales. An equivalent analysis could not be easily accomplished with single-unit activity. Our main goal was to use whatever brain signal allows us to investigate the temporal dynamics of neural timescales in this

unconstrained foraging context from a population perspective. Notably, even timescales estimated from neuronal spiking have been generally described and reported as a population-level statistic due to their high heterogeneity exhibited at the single-neuron level (see Murray et al., 2014). That being said, although LFP and single-unit activity are fundamentally different signals, Gao et al., 2020 showed that macaque ECoG timescales track previously published spiking timescales (Murray et al., 2014). Moreover, we have previously shown that neural timescales estimated from neuronal spiking and fMRI, a modality closely related to LFP (Logothetis et al., 2001), display a very similar hierarchical organization (Manea et al., 2022). There is also evidence that the unimodal areas to association areas axis of hierarchical organization of neural timescales is preserved in humans (Ito et al., 2020; Raut et al., 2020). Given all this work, we argue that there is an overlap in how neural timescales at rest are organized across modalities and species. Nevertheless, the exact correspondence between neuronal spiking, LFP and BOLD timescales and the extent of their overlap remains unclear. Likewise, the dynamics of neural timescales across modalities and species remain an empirical question.

R.2.2. Comment: I also felt the abstract and introduction were long and convoluted. Moreover, it is assumed the reader is familiar with the concepts of “hierarchies of neural timescales” from the very first sentence of the abstract. I strongly encourage taking a sentence or two to define this first. I would recommend shortening the intro and getting to the point of why this metric matters. I liked the 4th paragraph which motivates the research question but felt like the first 3 paragraphs could be condensed into one or two, making this an easier read.

R.2.2. Response: We thank the reviewer. We have added a definition and shortened the introduction as suggested. We have also included a definition at the beginning of the abstract as suggested stating “...*hierarchy of neural timescales at rest, with sensory areas displaying fast, and higher-order association areas displaying slower temporal characteristics.*”

R.2.3. Comment: I’m sorry to say that I did not feel an excitement for those results commensurate with the prestige of the journal they are submitted to. The use of a highly-derived, post-processed neural metric with unknown functional relevance in the context of a purely correlational paradigm did not strike my curiosity. This may simply reflect my ignorance towards this subject however, and other reviewers may beg to differ.

R.2.3. Response:

We acknowledge that the study of neural timescales in macaques is fairly recent and understanding their functional relevance is an ongoing effort. However, investigating how the brain handles multitudes of timescales has been extensively studied and the concept of “timescales” has been demonstrated to be functionally relevant for neural processing - for example, see work on temporal receptive windows in humans, or the temporal windows over which the brain integrates information (for example, see Hasson et al., 2008), timescales of population coding in macaques (for example, see Runyan et al., 2017), contextual modulation in macaques (Zimmermann et al., 2018). Moreover, work from human psychiatric populations brings evidence for the functional relevance of intrinsic neural timescales (Watanabe et al., 2019; Wengler et al., 2020; Xie et al., 2023; Zilio et al., 2021). We believe that our results are not only important for understanding neural timescales in the context of behavior, but also crucial for translational purposes.

The freely moving paradigm is indeed unstructured which makes it perfect for studying neural timescales since we are imposing relatively minimal temporal structure. However, this also comes with disadvantages as the reviewer has pointed out, since it is not trivial to establish causal relationships given current technologies used in nonhuman primates. We argue that our paradigm is complementary

to more structured paradigms, and it is vitally important to investigate neural processing in a naturalistic environment, providing ecological validity.

Revised Discussion (page 21):

Nevertheless, the specific meaning of these time-locked local changes in the magnitude of neural timescales remains an outstanding question. Although ecologically valid, unconstrained behavior automatically introduces variability that cannot be controlled for, nor is easily modeled. Our results therefore could be extended by directly manipulating the information integration across multiple temporal scales, while introducing clear task endpoints.

Reviewer #3:

R3.1. Comment: In the introduction, the authors claim “There is currently conflicting evidence about the behavioral dependence of neural timescales, their hierarchical organization and general function” and as evidence for this claim, they refer to studies that show timescales can change with task demands and compare them with studies that found a hierarchical organization of timescales across the brain at rest. However, I do not see any controversy between these two sets of findings. Having a replicable hierarchical organization at rest does not exclude the possibility that timescales can change during task performance. In fact, Gao et al., 2020 provided evidence for the existence of both phenomena in the same data: the hierarchical organization of timescales within association regions exists during the baseline period and all regions exhibit a ~20% increase in their timescales after engagement in the working memory task.

R3.1. Response: We agree with the reviewer in that the phrasing of controversy may be misleading but a recent review (Cavanagh, 2020) has reemphasized that the stationary assumptions on intrinsic timescales have to be re-evaluated in light of the heterogeneity of neural timescales at the single-neuron level. We believe that the disagreement relates to the intersection of hierarchical organization and behavioral relevance.

We indeed cited Gao et al., 2020 and Zeraati et al., 2023 as examples of studies providing evidence that neural timescales expand with task engagement. It is important to note that the hierarchical organization found by Gao et al., 2020 does not match previous work. In particular, their OFC timescales are the slowest when compared to those in more dorsal PFC regions, which is the opposite of what we usually see with neural timescales estimated from monkey spiking data at rest. In particular, we argue that neural timescales are functionally relevant, and hence this variability across studies could be mainly due to different task contexts and behavioral demands. To eliminate any ambiguities, we have thus rephrased the introduction accordingly.

Revised Introduction (page 3):

“There is currently scarce evidence about the behavioral dependence of neural timescales, their hierarchical organization and general function in the context of behavior, and especially how these relate to one another.”

R3.2. Comment: In the introduction, the authors claim “ We propose that the failure to find multiple hierarchies of neural timescales in previous experiments may be a by-product of the rigid structure of traditional experiments that enforce stationary temporal scales.”, but it is not clear what they mean by “multiple hierarchies”. Previous studies such as Spitmaan et al., 2020 have found multiple hierarchies of timescales (i.e. intrinsic and task-related timescales) that all follow a similar order of brain areas. Authors might mean that all previous studies suggested a similar order of brain areas for the hierarchy, in that case, it is better to clarify.

R3.2. Response: We thank the reviewer for pointing out this lack of clarity and have changed the introduction to reflect this.

Revised Introduction (page 4):

We propose that the failure to find multiple hierarchies of neural timescales in previous experiments, as reflected by the areas' relative position, may be a by-product of the rigid structure of traditional experiments that enforce stationary temporal scales.

R3.3. Comment: For readers to better contextualize the results, I think it is better to unpack what the authors mean by “In summary, we found a ventro-dorsal hierarchy that partially overlapped but also deviated from previous findings in important ways” in the results section. Detailed explanations for this difference are now provided in the discussion, but having some sort of visualization that compares the order of brain areas in the previous studies and the current study is helpful in understanding the results.

R3.3. Response: We thank the reviewer and we added overview **Fig. 6**, in particular Panel A which places the hierarchy reported here in the context of neural timescales estimated across different studies, species and modalities for a better contextualization of the results.

R3.4. Comment: For measuring timescales authors consider two types of windows: (i) Hamming window of size 1 sec for computing PSD, (ii) Sliding window of size 10/5 sec (depending on the analysis) to compute time-dependent timescales. It is important to show that the results presented in the paper are not dependent on the chosen window sizes, or at least show within what range of window sizes the results are replicable. This is particularly important since the size of Hamming window defines the largest estimated timescale and the size of the timescale window (if too larger) might hinder some changes in the neural timescales. It is also not clear why the authors chose a different window size for the analysis in Fig 2A, compared to the rest of the paper.

R3.4. Response: We thank the reviewer for this comment, and we have provided an additional supplementary figure that shows that the relative order of the brain areas does not change for Hamming windows of size 500, 1000 and 1500 ms (see **Supplementary Fig. 5**). We demonstrate that this is the case in a subset of the data (i.e., 10 sessions in monkey W and 10 sessions in Monkey Y). Unfortunately, the analysis pipeline is very computationally expensive which makes it very difficult to replicate all results using a range of Hamming window sizes.

Generally, the timescales window size is a way to increase signal to noise ratio when estimating the neural timescales. For the session-wide timescales, we used a larger window size (i.e., 10 s) since we were interested in the general trend, their hierarchical organization over time and relationship to our task engagement index. Given the sparsity of the lever presses, we had to downsample the data by looking at 10-min segments. Lastly, a larger window size reduces the amount of fitted models, and hence makes the analysis less computationally expensive. Given that a higher temporal resolution was not necessary for this analysis, we decided to use the 10s window. For the event-related timescales, we chose a window size half of that as for the session wide timescales in order to increase temporal fidelity while balancing computational feasibility as well as stability of the power spectral density estimates. The reviewer makes a good point that the size of the timescale window determines the level of detail we uncover. For finer dynamics, especially when investigating dynamic transitions between behavioral repertoires (as in trialized chaired experiments) smaller window sizes should be used but our hypotheses were best addressed using the given parameters.

Revised Methods (page 25):

*To examine whether the Hamming window size has an effect on the areas' relative position in the hierarchy, we also estimated the PSDs using 500 ms (250 ms overlap) and 1500 ms (750 ms overlap) for 20 recording sessions where data was available for all areas (Monkey W: 10; Monkey Y: 10). We did not find any evidence for the Hamming window size having an impact on the areas' relative position in the hierarchy (see **Supplementary Fig. 5**).*

R3.5. Comment: Regarding “To assess the significance of this linear increase for individual areas, we conducted a linear regression model for each event with time as the predictor and neural timescales as the dependent variable.” (page 18), details of the analysis are unclear and do not quite match the visualizations in Fig 3 C, D. The regression analysis should potentially be done using data around all individual events (across sessions/monkeys) and not the summary of data (depending on the variability across sessions, separate intercepts might need to be fitted). However, in Fig 3 C, D, it seems some summary of data (unclear whether mean/median) without any error bars are presented. A proper visualization for the regression analysis, would be to plot all the measured timescales (around each event across sessions/monkeys) relative to the time to/from the event together with the fitted regression line. Moreover, the parameters of regression analysis (e.g., the slope), number of data samples and R-squared need to be reported (e.g., in a Table), since significance alone might not indicate strong correlations between timescales and their relative temporal distance to the events.

R3.5. Response: We thank the reviewer and agree with the importance of a more detailed description of the methods. We have indeed fit a linear regression model for each recording session and area. We have now added **Supplementary Fig. 4** which depicts the distributions of the resulting standardized regression coefficients per area (and includes the average R^2 across sessions). Given that we fit a regression model for each area and recording session, a visualization of all the fitted regression lines would not be informative. We believe that the figure depicting the distributions of standardized regression coefficients is the most informative graphical depiction of the analysis.

R3.6. Comment: The amount of change in the magnitude of timescale across brain areas is computed as the difference to the baseline (task-free) but this comparison is more meaningful if the difference is also normalized to the baseline of each area, i.e. $(\tau(\text{task}) - \tau(\text{baseline})) / \tau(\text{baseline})$. This relative normalization ensures that the structure presented in Fig 4, 5 is not a mere consequence of brain areas with larger timescales having a larger absolute change in their timescales and hence providing no additional insight to the already observed hierarchy in the magnitude of timescales (for instance, it might be the case that all areas are increasing their timescales by the same percentage suggesting that a global mechanism is modulating the timescales across all areas).

R3.6. Response: We thank the reviewer for this comment and question. Based on the results in **Fig. 4**, we conclude that there are both global changes and more targeted changes related to foraging events. For example, in Fig. 4A-B, before the event at time point -5, there seems to be a global absolute change with little differentiation between areas (i.e., most areas are not significantly different from each other). However, at time point -1, we see a clear statistical differentiation between areas, with only PM-SMA-FEF, DLPFC-VLPFC-ACC and OFC-Striatum displaying similar change. Our interpretation is that a global mechanism drives the changes from baseline, with the exception of time points in proximity of the events of interest.

Additionally, we provide a figure for reviewers that reflects the change as percentage of baseline. A global mechanism would indeed be reflected in an increase by the same baseline percentage across all areas at any time point. However, this does not seem to be the case with the FEF displaying the highest change as percentage of baseline and the VLPFC, which is low in the hierarchy with respect to its absolute change, displaying a similar percentage change to more dorsal areas. There are still significant differences between the areas when that information is removed which excludes a global mechanism influencing the results. Finally, to have a better summary of the results in **Fig. 4**, we estimated the effect size for each timepoint and plotted it in **Fig. 6B**. As stated above, the differences between areas are not stationary but rather change across time. If the results in **Fig. 4** would be solely driven by slower areas displaying higher absolute changes from baseline, the effect size should not vary across time. We conclude that both the areas' baseline and area-specific responses to the events are driving the results in **Fig. 4**.

Figure for reviewers. Area-specific adaptation of neural timescales represented as percentage of baseline. The change was calculated as the difference between event-related neural timescales (first and final lever presses) and resting-baseline divided by the resting-baseline. Circles: median across sessions \pm s.e.m.

Revised Results (page 13):

To quantify these two types of observed changes (i.e., global and area-specific changes), for each time point, we calculated the effect size of each Mann-Whitney U-test by estimating the rank-biserial correlation coefficient (Wendt, 1972). Next, for each time point we averaged the absolute value of the effect sizes across the Mann-Whitney U-tests (i.e., associated with each time point's 28 pairwise comparisons). For each time point, the average rank-biserial correlation coefficient was used as an index of how differentiated the areas were in their change from baseline. This index ranges from 0, when all areas display the exact same change from baseline, to 1, which would indicate complete independence in their change from baseline. Supporting our previous conclusion, the differentiation between areas increased in the moments leading up to the lever presses (see overview **Fig. 6B**). However, the two types of events (first and final lever presses) diverge after the event. In particular, the first lever presses display a drop, while the final lever presses display a continuation of this differentiation between areas. We conclude that there are both global changes, potentially driven by

similar mechanisms across areas, but also more targeted effects that differentiate the areas in their response to foraging events.

R3.7. Comment: It is difficult to draw clear conclusions from all the changes presented in Fig 4, 5, providing a summary figure with the main conclusions (considering only statistically significant results) would be helpful.

R3.7. Response: We thank the reviewer, and we agree that an overview figure makes it easier to see the bigger picture. We added **Fig. 6** which places our hierarchy in the context of other studies/species/modalities (panel A), provides a summary statistic for **Fig. 4** (panel B) and a schematic of the main conclusions of **Fig. 5** (panel C).

R3.8. Comment: Minor comments and clarifications:

- Please add line numbers to the manuscript.
- It seems the authors are using the phrases “segments/bins” interchangeably (e.g., Fig 2C), it would be nice to make it consistent across the text using one of the two phrases. Please also mention the size of the bin in the caption of Fig 2C, D.
- In the caption of Fig 3B-D and Fig 4B, C, please mention what dots (and error bars if plotted) indicate (mean/median/etc), and how are they computed (across sessions/monkeys/etc). Please also explain what the dashed circles represent in Fig 4.
- For figures that you computed p-values, please add the number of samples to the caption and for Bonferroni corrections please mention for how many multiple comparisons.

R3.8. Response: All minor comments have been addressed in this revision, thank you.

References:

- Gao, R., van den Brink, R. L., Pfeffer, T., & Voytek, B. (2020). Neuronal timescales are functionally dynamic and shaped by cortical microarchitecture. *ELife*, *9*, e61277. <https://doi.org/10.7554/eLife.61277>
- Hasson, U., Yang, E., Vallines, I., Heeger, D. J., & Rubin, N. (2008). A Hierarchy of Temporal Receptive Windows in Human Cortex. *The Journal of Neuroscience*, *28*(10), 2539. <https://doi.org/10.1523/JNEUROSCI.5487-07.2008>
- Ito, T., Hearne, L. J., & Cole, M. W. (2020). A cortical hierarchy of localized and distributed processes revealed via dissociation of task activations, connectivity changes, and intrinsic timescales. *NeuroImage*, *221*, 117141. <https://doi.org/10.1016/j.neuroimage.2020.117141>
- Logothetis, N. K., Pauls, J., Augath, M., Trinath, T., & Oeltermann, A. (2001). Neurophysiological investigation of the basis of the fMRI signal. *Nature*, *412*(6843), 150–157. <https://doi.org/10.1038/35084005>
- Manea, A. M., Zilverstand, A., Ugurbil, K., Heilbronner, S. R., & Zimmermann, J. (2022). Intrinsic timescales as an organizational principle of neural processing across the whole rhesus macaque brain. *ELife*, *11*, e75540. <https://doi.org/10.7554/eLife.75540>
- Murray, J. D., Bernacchia, A., Freedman, D. J., Romo, R., Wallis, J. D., Cai, X., Padoa-Schioppa, C., Pasternak, T., Seo, H., Lee, D., & Wang, X.-J. (2014). A hierarchy of intrinsic timescales across primate cortex. *Nature Neuroscience*, *17*(12), Article 12. <https://doi.org/10.1038/nn.3862>
- Raut, R. V., Snyder, A. Z., & Raichle, M. E. (2020). Hierarchical dynamics as a macroscopic organizing principle of the human brain. *Proceedings of the National Academy of Sciences*, *117*(34), 20890–20897. <https://doi.org/10.1073/pnas.2003383117>
- Runyan, C. A., Piasini, E., Panzeri, S., & Harvey, C. D. (2017). Distinct timescales of population coding across cortex. *Nature*, *548*(7665), 92–96. <https://doi.org/10.1038/nature23020>
- Watanabe, T., Rees, G., & Masuda, N. (2019). Atypical intrinsic neural timescale in autism. *ELife*, *8*, e42256.
- Wengler, K., Goldberg, A. T., Chahine, G., & Horga, G. (2020). Distinct hierarchical alterations of intrinsic neural timescales account for different manifestations of psychosis. *ELife*, *9*, e56151.

- Xie, K., Royer, J., Lariviere, S., Rodriguez-Cruces, R., de Wael, R. V., Park, B., Auer, H., Tavakol, S., DeKraker, J., & Abdallah, C. (2023). Atypical intrinsic neural timescales in temporal lobe epilepsy. *Epilepsia*, 64(4), 998–1011.
- Zeraati, R., Shi, Y.-L., Steinmetz, N. A., Gieselmann, M. A., Thiele, A., Moore, T., Levina, A., & Engel, T. A. (2023). Intrinsic timescales in the visual cortex change with selective attention and reflect spatial connectivity. *Nature Communications*, 14(1), 1858. <https://doi.org/10.1038/s41467-023-37613-7>
- Zilio, F., Gomez-Pilar, J., Cao, S., Zhang, J., Zang, D., Qi, Z., Tan, J., Hiromi, T., Wu, X., Fogel, S., Huang, Z., Hohmann, M. R., Fomina, T., Synofzik, M., Grosse-Wentrup, M., Owen, A. M., & Northoff, G. (2021). Are intrinsic neural timescales related to sensory processing? Evidence from abnormal behavioral states. *NeuroImage*, 226, 117579. <https://doi.org/10.1016/j.neuroimage.2020.117579>
- Zimmermann, J., Glimcher, P. W., & Louie, K. (2018). Multiple timescales of normalized value coding underlie adaptive choice behavior. *Nature Communications*, 9(1), 3206. <https://doi.org/10.1038/s41467-018-05507-8>

REVIEWERS' COMMENTS

Reviewer #1 (Remarks to the Author):

I have read the authors response to my original review comments, as well as the response to the other reviewer, and checked the updated manuscript. Overall, I find that the response and updates to the manuscript to address my main suggestions for the manuscript. I would like to thank the reviewers for the significant updates to the manuscript, which I think have improved the manuscript. I have no further comments or suggestions that I think should need to be addressed before this manuscript is ready for publication.

Reviewer #2 (Remarks to the Author):

I praise the authors for attempting to answer to the numerous comments raised by the three reviewers. Unfortunately for the authors, I did the annoying thing of reading the comments from the other reviewers. I must praise their hard work too.

I couldn't help but notice that Reviewer 1 and myself share the same major concern about the comparison of data in the current study with the rest of the literature. More precisely, we see issues contrasting neural timescales computed on spikes with neural timescales computed on LFPs. The authors did a significant job trying to address these concerns mainly by stating the following argument: the same hierarchy is found using spikes, ECoG and fMRI at rest, hence, the different modality used herein (i.e. LFP) cannot explain the differences observed in timescales. A new figure, Figure 6, helps to illustrate this point.

I am sorry to say that despite this argument, I remain unconvinced. When I compare the four subplots on the right of Fig 6A, in all honesty, I see that results are all over the place. I do not find replicable trends in hierarchies there across modalities, even if I blink hard. Hence the argument made above by the authors appears untenable.

I still believe that there are too many possible alternative explanations for the differences observed in the current study compared to the rest of the literature, namely the LFP metric used. To me, it appears that hierarchies are modality-dependent, and moreover, dependent on the numerous analytical factors enumerated by reviewer 3 that also have an impact on results, which brings me to my last point, which is more philosophical in nature.

The power of animal research, and monkey research in particular, is the unrivalled ability to capture **meaningful** biological signals to understand the function of organs, including the brain. We have learned so much from techniques like intracranial neurophysiology, in part because we have been able to record a clear, meaningful biological signal from the brain of primates: action potentials. It is a simple metric with clear biological relevance that requires minimal pre-processing to capture in the context of behavior.

In comparison, the current paper is based on the following metric: "*the hierarchy of the timescales of the aperiodic component fit of the power spectral density of the local field potentials*". I hope you see the point. This highly derived metric, which can be computed with infinite "experimenter's degrees of freedom", has a long way to go before convincing a biologist like myself that it matters at all.

I could be convinced that a highly-derived metric is a useful "proxy" if its signal-to-noise ratio is very high and it varies clearly along some meaningful biological axis. However, looking at Fig 1D, I see neither of that. I see a highly noisy signal from "example channels" (which are usually not the worst) with a thresholding operation (dashed vertical lines) that appears also very noisy. Looking at those coloured lines, I see "knees" everywhere I want to see one. And because the "hierarchy" is based on an ordinal order, a small difference in the position of a "knee" by a few Hz changes the hierarchy drastically, and hence the conclusions of this article. In brief, I think this metric is highly

problematic, which has serious impacts of the value of this work.

In conclusion, I praise the authors for a truly fantastic piece of engineering work. I am a big fan of this research group and I wish them the best. However, I am unconvinced about the value of this work from a strictly biological perspective. Once again, I am not an engineer and this may reflect my own biases. I am sure many smart people would disagree with my point of view.

Best of luck.

Reviewer #3 (Remarks to the Author):

I thank the authors for a comprehensive rebuttal and revision of the manuscript, which has made the presentation of results more clear, strengthened the manuscript, and addressed my major concerns. I just have some minor comments that can be addressed in a revision.

Minor comments:

- Comparison to Gao et al., 2020 timescales:

For Fig 6A comparisons, the authors included the human LFP results from Gao et al., 2020, but in the same paper (Fig 1F), there are also results for Monkey LFP (ECoG) timescales which are a better comparison to the results in this manuscript and have a larger coverage of areas. So, I recommend adding that comparison to Fig 6A and also where relevant to the Discussion when mentioning the results from this paper.

- Figure for reviewers (changes in timescales as a percentage of baseline):

I recommend adding this figure to the supplementary since it is informative for the readers and helpful for future extensions of this work. It also supports the claim that there are indeed some area-specific changes even after normalizing with baseline.

- Fig 4:

For panels B,C: please clarify in the caption what the dots represent: are they mean/median across sessions or data from an example session?

- Fig 3. (and other similar figures):

If s.e.m refers to the standard error of the mean, then, plotting the median +/- s.e.m is not statistically informative. The s.e.m describes the error bars relative to the mean of the distribution, and not to the median. For error bars relative to the median, it is more informative to plot/report the percentiles (e.g., 25% and 75%). Alternatively, authors can also plot/report the mean +/- s.e.m.

Reviewer 1:

I have read the authors response to my original review comments, as well as the response to the other reviewer, and checked the updated manuscript. Overall, I find that the response and updates to the manuscript to address my main suggestions for the manuscript. I would like to thank the reviewers for the significant updates to the manuscript, which I think have improved the manuscript. I have no further comments or suggestions that I think should need to be addressed before this manuscript is ready for publication.

We thank the reviewer for their contribution. The comments and suggestions have significantly improved the manuscript.

Reviewer 2:

I praise the authors for attempting to answer to the numerous comments raised by the three reviewers. Unfortunately for the authors, I did the annoying thing of reading the comments from the other reviewers. I must praise their hard work too.

I couldn't help but notice that Reviewer 1 and myself share the same major concern about the comparison of data in the current study with the rest of the literature. More precisely, we see issues contrasting neural timescales computed on spikes with neural timescales computed on LFPs. The authors did a significant job trying to address these concerns mainly by stating the following argument: the same hierarchy is found using spikes, ECoG and fMRI at rest, hence, the different modality used herein (i.e. LFP) cannot explain the differences observed in timescales. A new figure, Figure 6, helps to illustrate this point.

I am sorry to say that despite this argument, I remain unconvinced. When I compare the four subplots on the right of Fig 6A, in all honesty, I see that results are all over the place. I do not find replicable trends in hierarchies there across modalities, even if I blink hard. Hence the argument made above by the authors appears untenable.

I still believe that there are too many possible alternative explanations for the differences observed in the current study compared to the rest of the literature, namely the LFP metric used. To me, it appears that hierarchies are modality-dependent, and moreover, dependent on the numerous analytical factors enumerated by reviewer 3 that also have an impact on results, which brings me to my last point, which is more philosophical in nature.

The power of animal research, and monkey research in particular, is the unrivalled ability to capture **meaningful** biological signals to understand the function of organs, including the brain. We have learned so much from techniques like intracranial neurophysiology, in part because we have been able to record a clear, meaningful biological signal from the brain of primates: action potentials. It is a simple metric with clear biological relevance that requires minimal pre-processing to capture in the context of behavior.

In comparison, the current paper is based on the following metric: *"the hierarchy of the timescales of the aperiodic component fit of the power spectral density of the local field potentials"*. I hope you see the point. This highly derived metric, which can be computed with infinite "experimenter's degrees of freedom", has a long way to go before convincing a biologist like myself that it matters at all.

I could be convinced that a highly-derived metric is a useful "proxy" if its signal-to-noise ratio is very high and it varies clearly along some meaningful biological axis. However, looking at Fig 1D, I see neither of that. I see a highly noisy signal from "example channels" (which are usually not the worst) with a thresholding operation

(dashed vertical lines) that appears also very noisy. Looking at those coloured lines, I see “knees” everywhere I want to see one. And because the “hierarchy” is based on an ordinal order, a small difference in the position of a “knee” by a few Hz changes the hierarchy drastically, and hence the conclusions of this article. In brief, I think this metric is highly problematic, which has serious impacts of the value of this work.

In conclusion, I praise the authors for a truly fantastic piece of engineering work. I am a big fan of this research group and I wish them the best. However, I am unconvinced about the value of this work from a strictly biological perspective. Once again, I am not an engineer and this may reflect my own biases. I am sure many smart people would disagree with my point of view.

Best of luck.

We thank the reviewer for this comment. We agree with the reviewer that the hierarchies are not entirely replicable even within the same modality. We believe that this is the case because neural timescales are heavily dependent on behavioral demands and will change depending on the task the animal is performing. We don't however agree with the metric in itself being highly problematic. It's a proxy, there is no arguing around that and we agree but proxies can be a good thing and we can learn a lot from them going forward. Here we are laying fundamental groundwork that needs to be done to go forward. We also agree with the reviewer that much validation work on said metrics needs to be done and the authors would argue that basic causal experiments are needed to that end. BOLD signals for example were heavily criticized (and rightfully so, still are in some domains) but they have undoubtedly been very influential and impactful from a biological perspective. Engineering, physics and neurobiology go hand in hand and progress is an incremental walk we do together. We are happy the author can see the value from an “engineering” perspective, and we hope to continue to work towards convincing the author of the biological value of timescales derived metrics going forward.

Reviewer 3:

I thank the authors for a comprehensive rebuttal and revision of the manuscript, which has made the presentation of results more clear, strengthened the manuscript, and addressed my major concerns. I just have some minor comments that can be addressed in a revision.

Minor comments:

R3.1 Comment: Comparison to Gao et al., 2020 timescales:

For Fig 6A comparisons, the authors included the human LFP results from Gao et al., 2020, but in the same paper (Fig 1F), there are also results for Monkey LFP (ECoG) timescales which are a better comparison to the results in this manuscript and have a larger coverage of areas. So, I recommend adding that comparison to Fig 6A and also where relevant to the Discussion when mentioning the results from this paper.

R3.2. Comment: Figure for reviewers (changes in timescales as a percentage of baseline):

I recommend adding this figure to the supplementary since it is informative for the readers and helpful for future extensions of this work. It also supports the claim that there are indeed some area-specific changes even after normalizing with baseline.

R3.3. Comment: Fig 4: For panels B,C: please clarify in the caption what the dots represent: are they mean/median across sessions or data from an example session?

R3.4. Comment: Fig 3. (and other similar figures): If s.e.m refers to the standard error of the mean, then, plotting the median +/- s.e.m is not statistically informative. The s.e.m describes the error bars relative to the mean of the distribution, and not to the median. For error bars relative to the median, it is more informative to plot/report the percentiles (e.g., 25% and 75%). Alternatively, authors can also plot/report the mean +/- s.e.m.

We thank the reviewer for their contribution. The comments and suggestions have significantly improved the manuscript.

R3.1. response:

We added the monkey ECoG results to Figure 6A and we changed the discussion accordingly.

R3.2. response:

We added Supplementary Figure 5 (changes in timescales as a percentage of baseline).

R3.3. response:

We clarified what the dots represent in Figure 4.

R3.4. response:

We agree that reporting the mean and SEM is the most statistically informative. However, the statistical tests employed in this manuscript use the median rather than the mean - hence, we believe that visualizing the median is more informative. For error bars relative to the median, plotting the percentiles would not be feasible with so many areas/timepoints since it would be difficult to visualize. To address this comment, we report the mean along with the mean and SE in the Source Files associated with all the figures.